# SPACT18: Spiking Human Action Recognition Benchmark Dataset with Complementary RGB and Thermal Modalities

## Abstract

Spike cameras, bio-inspired vision sensors, asynchronously fire spikes by accumulating light intensities at each pixel, offering ultra-high energy efficiency and exceptional temporal resolution. Unlike event cameras, which record changes in light intensity to capture motion, spike cameras provide even finer spatiotemporal resolution and a more precise representation of continuous changes. In this paper, we introduce the first video action recognition (VAR) dataset using spike camera, alongside synchronized RGB and thermal modalities, to enable comprehensive benchmarking for Spiking Neural Networks (SNNs). By preserving the inherent sparsity and temporal precision of spiking data, our three datasets offer a unique platform for exploring multimodal video understanding and serve as a valuable resource for directly comparing spiking, thermal, and RGB modalities. This work contributes a novel dataset that will drive research in energy-efficient, ultra-low-power video understanding, specifically for action recognition tasks using spike-based data.

## 1 Introduction

Video Action Recognition (VAR) is a key task in computer vision that focuses on detecting and classifying actions or activities from video sequences automatically (Wani & Faridi, 2022). Unlike image classification, which can be seen as analysis of an individual frame, VAR captures both spatial and temporal dynamics, adding complexity such as heavy information redundancy introduced by the temporal dimension, motion variations, changes in lighting or camera angles (Yu et al., 2024), just to name a few. These complexities make VAR highly valuable for real-world applications such as surveillance, healthcare, industrial automation, and sports analysis (Poppe, 2010), where rapid and accurate processing is essential.

Spiking Neural Networks (SNNs), inspired by biological neurons, offer a promising alternative to traditional Artificial Neural Networks (ANNs) for tasks involving temporal dynamics, such as VAR (Guo et al., 2023). SNNs operate on sparse, discrete spikes, enabling event-driven computation, which significantly reduces energy consumption compared to ANNs (Roy et al., 2019). This makes SNNs particularly suited for energy-constrained environments like autonomous robots and edge devices (Liu et al., 2024). However, despite their energy efficiency, video understanding with SNNs remains limited. Current research on VAR using SNNs typically relies on converting conventional RGB video data into spike trains, leading to information loss and constraining the full potential of SNNs (Yu et al., 2024).

A key limitation in existing research is the scarcity of spiking datasets that capture the rich temporal dynamics of real-world video sequences. While event cameras capture data based on brightness changes, spike cameras operate by generating spikes when the photon accumulation at each pixel surpasses a threshold. This enables spike cameras to capture absolute brightness at ultra-high sampling frequencies (up to 20,000 Hz), providing textural spatiotemporal details than event cameras (Amir et al., 2021). Despite these advantages, existing datasets, such as DVS128 (see Section 2.1), are based on event cameras and lack the richness needed for complex action recognition tasks.

In the context of SNNs, video data presents unique challenges. Videos inherently involve temporal sequences, which align naturally with the temporal nature of SNNs. Any temporally structured data

can be treated as a video, provided that each time step has a corresponding visual representation. When processing temporal input with an SNN, two distinct approaches arise: (1) aligning the temporal dimension of the input with the native temporal axis of the SNN model, or (2) encoding the entire video input separately from the SNN's native temporal axis. In the latter approach, the video is encoded as spike train as a whole, hence another temporal dimension is added which coincides with the SNN's temporal axis.

Motivated by the lack of a complex spiking dataset suitable to challenge and push forward the SNNs capacity to address VAR task, we introduce a novel dataset specifically designed for human action recognition using spike cameras. In addition, we provide spiking data with synchronized RGB and thermal modalities to enhance motion capture in low-light conditioning and provide a comprehensive multimodal representation of human actions. By combining these modalities, we provide a framework to explore the complementary information across different sensor types, with the aim of improving the robustness and diversity of SNN models for action recognition tasks.

Our datasets are collected from 44 participants, representing a wide range of demographic factors such as age, height, weight, sex, and ethnicity. Each participant performed 18 distinct daily actions (Figure 1), captured over two sessions, resulting in a total dataset duration of 264 minutes. To establish a baseline for comparison, we utilized state-of-the-art (SOTA) architecture based on convolutional neural networks (CNNs) and transformer blocks to provide a baseline model for our datasets in both ANN and SNN through ANN-SNN conversion. To summarize, our key contributions are as follows:

- We introduce SPACT18, the first VAR dataset captured using spike camera, setting a new benchmark for SNN-based models, and extend it with synchronized RGB and thermal modalities for comprehensive multimodal benchmarking.
- We propose a compression algorithm for the spiking dataset, yielding new spiking datasets with lower native latency, while preserving critical temporal information, providing a framework for preprocessing and compression for the research community.
- We evaluate our dataset across modalities using SOTA lightweight ANN models and SNN baseline obtained through ANN-SNN conversion, and direct training, highlighting critical challenges in spiking video recognition, such as the high latency in ANN-SNN and low accuracy in direct training, and providing novel research areas for optimizing SNNs.

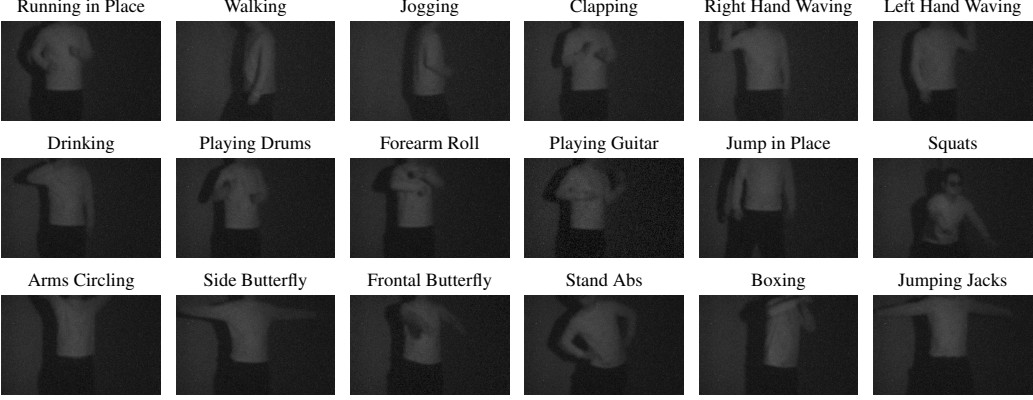

Figure 1: Sample output frame from the spike camera for each action of the same subject. Texture reconstruction via TFP (Dong et al., 2017) with window=200.

## 2 RELATED WORK

### 2.1 VIDEO ACTION RECOGNITION DATASETS

Video understanding has become a crucial component of computer vision, particularly due to the rise of short video platforms. To advance this field, datasets like KTH, simple human actions like walk-

ing and running (Schuldt et al., 2004), HMDB51, human actions in movies (Kuehne et al., 2011), UCF101, varied sports and action categories (Soomro et al., 2012), NTU RGB+D, multi-view human activities with depth data (Liu et al., 2020), and Kinetics, diverse real-world action videos (Carreira et al., 2018; 2022; Kay et al., 2017), have been developed for human action recognition, while others focus on action localization (Gao et al., 2017; Gu et al., 2018; Heilbron et al., 2015; Idrees et al., 2017; Liu et al., 2022a; Shao et al., 2020), enabling deeper exploration of human activities. Neuromorphic datasets (Wang et al., 2024b; Duan, 2024) like DVS128 (Amir et al., 2021), DailyDVS200 Wang et al. (2024a) along others for event-based action recognition (Dong et al., 2023; Gao et al., 2023) that utilize event-based cameras to capture spatiotemporal changes, recording only pixel intensity variations, which allows for efficient, real-time, low-power action recognition. Event-based versions of traditional datasets, such as E-KTH, E-UCF11, and E-HMDB51, are synthesized using event cameras or simulators, converting frame-based data into spike trains, which reduces redundancy and enhances processing efficiency, making these datasets ideal for applications requiring high temporal resolution (Bi et al., 2020; Al-Obaidi et al., 2021), see Appendix A.2 for more details.

## 2.2 SPIKE CAMERA DATASETS

Spike cameras have garnered attention for capturing high-speed dynamic scenes, spurring the creation of datasets to improve tasks like motion estimation, depth estimation, and image reconstruction. Zhu et al. (2020) introduced a dataset for reconstructing visual textures using a retina-inspired sampling method, demonstrating the utility of spike cameras for tasks like object tracking but limiting their scope to low-level applications. Similarly, datasets like Spk2ImgNet and PKU-Spike-Stereo by (Zhao et al., 2021) and (Wang et al., 2022b) focus on dynamic scene reconstruction and stereo depth estimation, respectively, without addressing high-level tasks such as video classification. Similarly, Hu et al. (2022) 's SCFlow dataset serves as a benchmark for optical flow estimation, and Zhao et al. (2020) developed a motion estimation dataset leveraging spike intervals for high-speed motion recovery, both confined to low-level vision tasks.

Despite these advancements, a significant gap remains in developing spike camera datasets for high-level tasks like video action recognition (Zhu et al., 2019; Xiang et al., 2023). Current datasets are predominantly designed for tasks such as optical flow (Zhao et al., 2022), motion estimation (Zheng et al., 2023), depth estimation (Zhang et al., 2022) and image reconstruction, limiting the exploration of spike camera data in vision understanding applications, such as event recognition and semantic understanding in dynamic environments. This gap underscores the need for future research on high-level spike-based vision tasks. Moving toward acceleration research in spiking video understanding, we introduce the first multimodal VAR dataset using a spike camera synchronized with RGB and thermal cameras, enhancing the understanding of VAR through complementary modalities.

## 2.3 VIDEO ACTION RECOGNITION ARCHITECTURES

Video understanding architectures have evolved from traditional CNNs to more sophisticated models. Early approaches like 3D CNNs, such as X3D Feichtenhofer (2020), extended 2D CNNs to capture both spatial and temporal aspects of videos Carreira & Zisserman (2017a); Tran et al. (2015; 2018a). Two-Stream Simonyan & Zisserman (2014), TSN Wang et al. (2016), and SlowFast Feichtenhofer et al. (2019b) further improved action recognition by incorporating spatial-temporal streams, sparse sampling, and parallel networks. Attention-based models, including TimeSformer Bertasius et al. (2021) and ViViT Arnab et al. (2021), enhanced temporal understanding by capturing long-range dependencies. Recent models, like VideoSwin Liu et al. (2022b) and UniFormer Li et al. (2022; 2023), combine convolution and self-attention for performance optimization.

On the other side, research on SNN-based video classification remains relatively limited. For instance, a reservoir recurrent SNN with 300-time-step spike sequences from UCF101 was proposed by Panda & Srinivasa (2018), while a heterogeneous recurrent SNN Chakraborty & Mukhopadhyay (2023) showed strong performance on datasets like UCF101, KTH, and DVS-Gesture. Wang et al. (2019) introduced the two-stream hybrid network (TSRNN), combining CNN, RNN, and a spiking module to enhance RNN memory. Another approach, a two-stream deep recurrent SNN, utilized ANN-to-SNN conversion, integrating channel-wise normalization and tandem learning Zhang et al. (2023a). To address conversion errors, You et al. (2024) developed a dual threshold mapping framework, reducing latency in SNNs using the SlowFast backbone. SVFormer, a spiking transformer

model, efficiently balances local feature extraction with global self-attention, achieving SOTA results with minimal power consumption Yu et al. (2024). Nonetheless, SNNs face challenges in complex model construction, preprocessing, and longer simulation times.

# 3 DATASET

To build a comprehensive and representative dataset, data were collected from 44 diverse subjects spanning a variety of cultural, gender, and racial backgrounds. The participants exhibited significant variability in key characteristics such as age, weight, height, lifestyle, and other factors, ensuring a robust and heterogeneous sample for our study, further enriching the dataset, and enhancing the model's ability to generalize across different demographics. Consent and approval have been taken from all participants to make the data publicly available for research purposes. Figure 2 summarizes an overview of the data collection process.

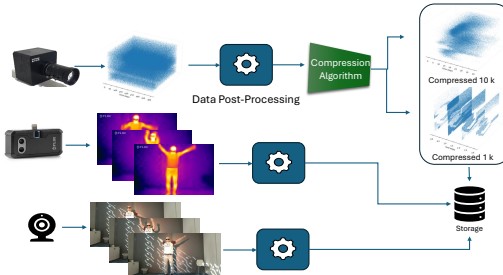

Figure 2: Multimodal data collection and compression pipeline: Spiking, Thermal, and RGB imaging with post-processing and compression techniques for optimized storage.

Table 1: Brief overview of the dataset.

Each participant conducted two separate data collection sessions, performing 18 distinct daily activities during each session, with each activity lasting 10 seconds. This protocol resulted in 180 seconds (3 minutes) of data per session, culminating in a total of 264 minutes ($\sim$ 4.5 hours) of data for each data type (spike, thermal, and RGB). Table 1 shows an overview of some statistical properties of the participants' numbers for the dataset.

| | |
|---|---|
| Height (cm) | 174.45 $\pm$ 8.23 |
| Weight (kg) | 75.41 $\pm$ 14.22 |
| Age (y) | 27.20 $\pm$ 5.11 |
| Participants | 44 |
| Nationalities | 13 |
| Activities | 18 |
| Sessions | 2 |
| Samples | 1584 |
| Sample Duration | 10 Seconds |

The activities are: *running in place*, *walking*, *jogging*, *clapping*, *waving with right hand*, *waving with left hand*, *drinking*, *playing drums*, *rolling with hands*, *playing guitar*, *jumping*, *squats*, *hand circling*, *side butterfly*, *front butterfly*, *standing ABS*, *boxing*, and *jump&jacks*. Sample output of the spike camera for each activity is shown in Figure 1. The selected activities encompass a broad spectrum of daily actions, deliberately chosen to include similar, closely related tasks and a mix of fast and slow-paced activities. This thoughtful selection allows the model to effectively learn and differentiate between actions, even those that are similar.

## 3.1 HARDWARE

As the data collection was conducted indoors, it was necessary to use a 2000-watt lamp to enhance the lighting conditions, thereby improving the image quality captured by the spike camera, which typically performs optimally under natural light. To ensure the safety of the participants, they were provided with sunglasses to protect their eyes from the direct and intense light. The data collection setup is shown in Figure 3. The three cameras were employed with the following specifications:

**Thermal Camera:** We utilized the FLIR One® Pro LT thermal camera, paired with a Samsung Galaxy S22 Ultra, to capture high-resolution thermal data. This compact, smartphone-compatible device ensured accurate temperature readings crucial for our study.

**RGB Camera:** We used the iPhone 14 Pro's rear camera to capture HD video at $1920 \times 1080$ resolution and 60 FPS, ensuring clear and fluid footage. Figure 4 presents a sample output from each camera.

**Spike camera:** is a novel imaging sensor designed to capture continuous, asynchronous signals by simulating the behavior of integrate-and-fire neurons. Spike cameras fire spikes for every pixel

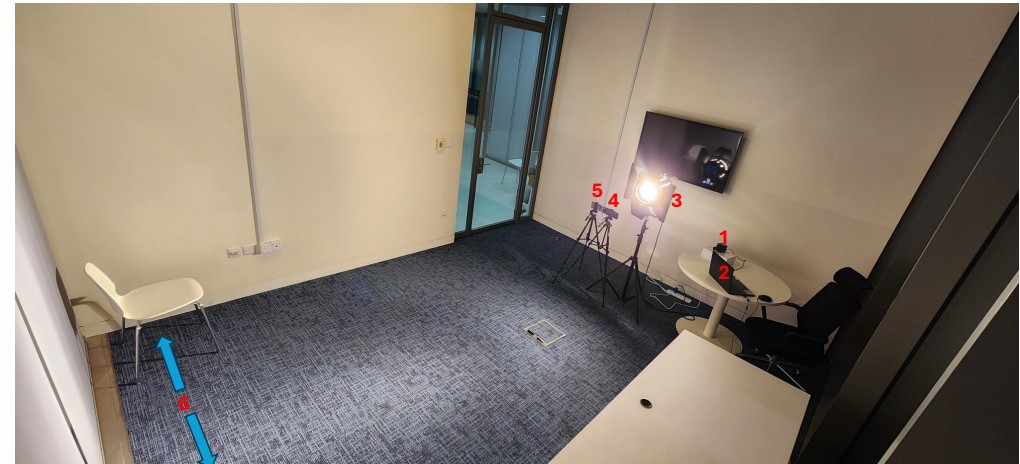

Figure 3: The experimental setup consists of the following components: (1) a spike camera for event-based visual sensing, (2) a laptop for real-time capture and recording of the spike camera output, (3) an artificial lighting source to ensure consistent illumination, (4) a thermal camera for infrared data collection, (5) RGB camera for standard color video recording, and (6) the designated area where subjects perform the activities under observation. The three cameras were fixed at a height of approximately 1 meter and placed 3.5 meters away from the subject performing the activities.

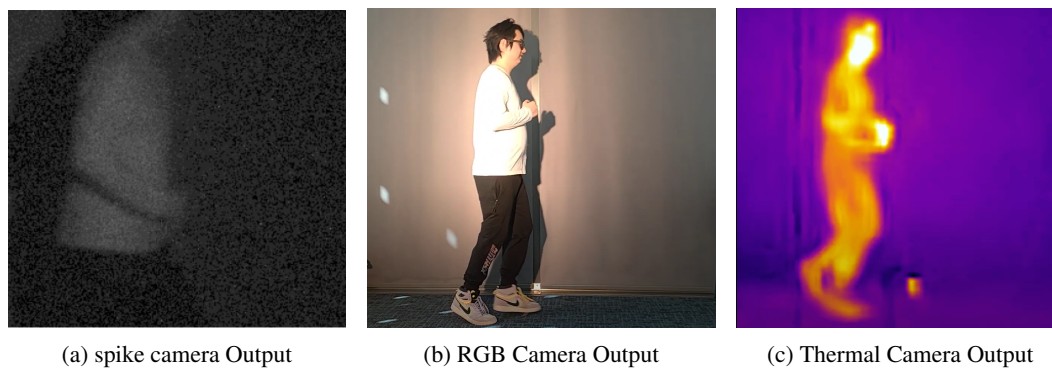

(a) spike camera Output          (b) RGB Camera Output          (c) Thermal Camera Output

Figure 4: Sample output frame from each camera for the same participant and action.

based on the intensity of incoming light. The core mechanism involves three key components per pixel: a photoreceptor, an integrator, and a comparator. The photoreceptor converts the scene's light into an electrical signal, which the integrator accumulates over time. Once the accumulated charge surpasses a predefined threshold $\theta$, the comparator triggers a spike and resets the integrator. This "integrate-and-fire" process occurs asynchronously for each pixel, producing a binary signal indicating whether a spike has been fired at any given time. The spike camera polls these signals at an extremely high frequency, generating spike frames that are arranged into a three-dimensional spike stream $S(x, y, k)$, where $x$ and $y$ represent pixel coordinates, and $k$ represents the discrete time index. Mathematically, the accumulation of the electric charge in the integrator can be represented by the following integral:

$$V_{i,j}(t) = \int_{t_{i,j}^{last}}^{t} \alpha \cdot I_{i,j}(\gamma)\, d\gamma$$

$$S(i, j, t) = \begin{cases} 1 & \text{if } V_{i,j}(t) \geq \theta, \\ 0 & \text{otherwise} \end{cases} \tag{1}$$

where $\alpha$ is the photoelectric conversion rate, $I_{i,j}(\gamma)$ is the light intensity at position $(i,j)$, $\theta$ is the firing threshold, and $t_{i,j}^{last}$ is last time when a spike is fired at position $(i,j)$. A spike is fired at time $t_k$ when the accumulated charge surpasses the threshold. At each polling interval $t = k\tau$, the system checks whether a spike has been triggered. Typically, for our camera we set $\tau = 50\mu s$. If the accumulated charge exceeds the firing threshold $\theta$, the spike is registered with $S(x,y,k) = 1$ and $V_{i,j}(t)$ is reset to zero; otherwise, $S(x,y,k) = 0$. These spike streams form a spatiotemporal binary matrix $\mathbf{M} \in \{0,1\}^{H \times W \times T}$ that captures the dynamic scene at high temporal resolution, enabling reconstruction of dense images from the sparse spike data.

## 3.2 DATA POST-PROCESSING

We implemented a rigorous data post-processing pipeline to guarantee the highest quality of the final dataset. Initially, we conducted a thorough manual inspection to identify and rectify any anomalies or errors that may have arisen during data collection, such as stops, doing the activities in the wrong order, or camera hardware issues. Subsequently, each session was segmented into 18 sub-videos, corresponding to individual activities. To further augment the dataset and introduce more significant variability, each sub-video was evenly divided into two additional sub-videos, culminating in a total of $3,168$ videos per data type. [1]

### 3.2.1 SPIKING DATA COMPRESSION

spike camera samples consist of a 100k-length binary spike trains, which poses significant challenges for SNN training and increases latency during inference. An established approach to overcome this and reduce the initial latency of the dataset is through temporal sampling of the spike camera data. However, using conventional sampling algorithms directly can result in aliasing issues and the loss of critical information due to the sparsity of spikes and the irregularity of spike occurrences.

---

**Algorithm 1** Spike Compression Algorithm

1: **Input:** $s, d$
2: **Initialize:** $u \leftarrow 0, T' \leftarrow \lfloor \frac{T}{d} \rfloor$
3: **for** $i = 0$ to $T' - 1$ **do**
4: $\quad r \leftarrow \text{mean}(s[i \cdot d + 1 : (i+1) \cdot d])$
5: $\quad v \leftarrow v + r$
6: $\quad s'[i] \leftarrow H(v - 1)$
7: $\quad v \leftarrow v - s'[i]$
8: **end for**
9: **return** $S_u$

---

In order to address long latency of the original spiking data (100k time steps), we propose a compression algorithm, designed to reduce the latency and at the same time to keep the original representation of the data. The main idea behind is to record the spiking rate at regular (of the same length), distinct (without overlapping) and exhaustive (the union covers the whole, or almost whole of 100k time steps) intervals, and produce the data with the same spiking rates, but recorder over the lower latency.

To say more, we put ourselves in a general situation and consider a spike train $s = [s[1], \ldots, s[T]]$ of length $T$ time steps. Let $d \ll T$ be the length of an interval and let $T' = \lfloor \frac{T}{d} \rfloor$. Then, our construction proceeds by considering $T'$ intervals (sets) of the form $I[i] := \{id + 1, \ldots, (i+1)d\}$, for $i = 0, \ldots, T' - 1$. Note that the union of the intervals covers $\{1, \ldots, T\}$, except for the last few elements of the form $dT' + 1, \ldots, T$ (in case $T$ is not divisible by $d$).

For each of the intervals $I[i]$, we define $r[i]$ to be the spiking rate of $s$ during the interval, i.e. $r[i] = \frac{1}{d} \sum_{t \in I[i]} s[t]$. Then, our compressed spike train $s'$ is obtained by using the Integrate-and-Fire (IF) spiking neuron which receives $r[i]$ as an input at the time step $i$, for $i = 0, \ldots, T' - 1$

$$v[i] = v[i-1] + r[i]$$
$$s'[i] = H(v[i] - 1) \tag{2}$$
$$v[i] = v[i] - s'[i].$$

---

[1]We provided the output of each stage for all data modalities on the shared drive, and it will be publicly available upon acceptance.

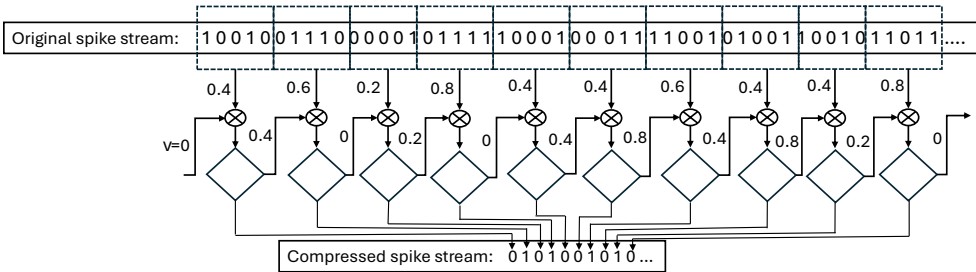

Figure 5: Illustration of the proposed spike stream compression method. The original spike stream is provided as input, and the compressed spike stream is produced as output with $T' = 5$. The membrane potential $u$ is initialized to zero and evolves as indicated in the diagram.

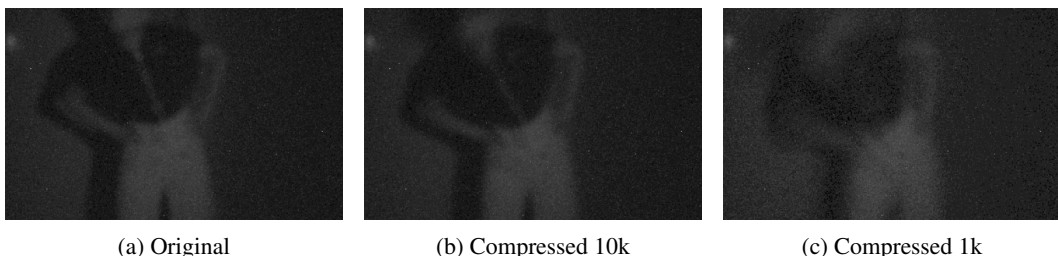

| (a) Original | (b) Compressed 10k | (c) Compressed 1k |

Figure 6: Reconstructed, with TFP, sample output frame from the spike camera for the three versions of our spiking datset original, compressed 10k, and compressed 1k.

We refer to Algorithm 1 for the algorithmic representation of the compression procedure, to Figure 5 for the graphical representation of the procedure and to Figure 6 for the comparisons of the reconstructed original and various compressed data.

Compression algorithm is motivated by the following result which demonstrates its soundness by explicitly showing the effect of compression on the special case of constant rate spike trains. We provide the proof and more details on terminology and setting in the supplementary material.

**Lemma 1.** *We keep the notation as above and suppose that $s$ is a spike train obtained when an IF neuron receives constant nonnegative input $c$ at each time step, and process it according to equations equation 2. Then, for any $d$ as above, the spike train $s'$ obtained through compression and spike train $s$ will have the same limit spiking rate (which is $c$).*

Using the above compression Algorithm 1, we have compressed the raw spiking data into two versions. We refer to them as compressed 10k and compressed 1k, with $d = 10$ (for 10k) and $d = 100$ (for 1k), respectively. These compressed versions were used for both training and testing.

Table 2: Comparison for storage of original and compressed spiking versions using LZMA.

| File | Original Size | Compressed Size (LZMA) |
|---|---|---|
| Original | 3.8 TB | 425 GB |
| Compressed 10k | 377.63 GB | 42.43 GB |
| Compressed 1k | 37.76 GB | 4.15 GB |
| Compressed 10k rate encoded | 240.76 GB | 80.96 MB |
| Compressed 1k rate encoded | 240.76 GB | 80.96 MB |

We need to further compress this data for efficient storage and transmission, as spiking videos are typically very large. However, the spike stream is a binary, sparse matrix, which makes it well-suited for compression. Table 2 shows the compression ratios achieved using the classic LZMA (Lempel-Ziv-Markov chain algorithm) (Pavlov, 2013), a lossless encoder, on the spike stream data.

## 4 EXPERMINTAL SETUP

To evaluate SPACT18, we adopted a systematic approach utilizing SOTA lightweight models. Given spiking data's energy efficiency and computational advantages, the results were benchmarked against highly efficient models to ensure fairness and relevance in future comparisons. We selected

two prominent lightweight architectures, X3D and UniFormer, known for their balance between computational efficiency and competitive accuracy on benchmark datasets.

X3D is a highly optimized CNN for video understanding Feichtenhofer (2020), balancing accuracy and efficiency by expanding input's temporal and spatial dimensions, reducing computational demands without sacrificing performance. This makes X3D particularly suitable for low-resource environments without compromising on accuracy. In contrast, UniFormer adopts a hybrid architecture that fuses convolutional operations with self-attention mechanisms Li et al. (2022). This allows the model to capture local and global dependencies in video data, making it versatile across different modalities. UniFormer's ability to effectively model spatiotemporal relationships with moderate computational cost ensures its competitive standing, even on challenging datasets.

We employed both small and medium variants for each model, which provide different trade-offs between model complexity and performance. The models were tested across all data modalities: thermal and RGB data (in both RGB and grayscale channels). Even though compressed 10k and 1k datasets significantly reduce the latency of the spiking dataset, training ANN models on them turned out to be infeasible. In order to provide baselines for them, we performed "rate" encoding of the datasets, which consists in the following. Instead of using the original binary 10k (resp. 1k) time steps for training the ANN, we reshaped the temporal dimension of spike trains into $100 \times 100$ (resp. $10 \times 100$) for compressed 10k (resp. 1k) dataset. We then computed the average along the first dimension to convert the binary spikes into rate-encoded values, effectively reducing the temporal complexity while preserving essential spike information for ANN training.

ANN-SNN conversion transforms a pre-trained ANN into a spike-based SNN (Diehl et al., 2015; Cao et al., 2015). For video models, this process is challenging due to the widespread use of non-ReLU activations and the depth of these models exacerbates the propagation of errors across layers. We selected MC3 to provide an SNN baseline for SPACT18. After training MC3, we recorded the maximum ReLU activations channel-wise using training dataset. This channel-wise approach ensures a more precise threshold for each neuron, improving the conversion accuracy. Then, each ReLU was replaced with a Leaky Integrate-and-Fire (LIF) neuron, using the recorded maximum activation as the neuron's threshold, while the initial membrane potential is set to half of the threshold. For inference, video inputs were encoded using a constant-encoding scheme.

The MC3 model is a variant of 3D CNNs for video action recognition, using a ResNet-18 backbone with 18 layers of 3D convolutions and two fully connected layers (Tran et al., 2018b). It employs a hybrid approach with 2D spatial and 1D temporal convolutions to reduce complexity while capturing motion information. Notably, MC3 utilizes ReLU activations, making it well-suited for ANN-SNN conversion and balancing accuracy and efficiency.

Table **??** presents the complete experimental setup, including all models and data modalities. The input size is defined as (Frame rate × Image dimension), and all experiments were conducted using a single RTX A6000 GPU. The dataset was then divided into 80% for training, 10% for validation, and 10% for testing. This split was performed subject-wise, ensuring that each subject and all corresponding activities remained within a single set, thus enabling a rigorous evaluation of the model's generalization capacity. Train, validation and test subjects are split at random and kept the same for all experiments for fair comparisons and consistency.

## 5 RESULTS AND DISCUSSION

The results, as shown in Table 3, show that thermal data consistently outperforms other modalities, allowing the model to focus solely on the activity and reducing distractions from irrelevant details. RGB color outperforms grayscale by adding richer information, although this advantage is more pronounced in RGB than in thermal data, where the additional channels have less impact.

For spiking data, the 10k compression level performed on par with thermal and RGB. However, performance dropped sharply at 1K compression, suggesting that spiking data retains critical information at higher compression levels but suffers significant loss at extreme compression. This highlights the challenge of processing spiking data compared to other modalities and underscores the need for selecting appropriate compression ratios. In terms of model efficiency, X3D is computationally lighter with lower FLOPs. Still, UniFormer consistently achieves higher accuracy, espe-

Table 3: Results for different experiments (RGB, Thermal, Spiking) for ANN models.

| Model | Data | Input Size | FLOPs (G) | Accuracy% | F1 Score |
|---|---|---|---|---|---|
| **X3D_M** *# Param = 3M* | RGB$_{Grey}$ | $50x224^2$ | $13.72\times3\times10$ | 78.7 | *0.76* |
| | RGB$_{Color}$ | $50x224^2$ | $41.19\times3\times10$ | 80.9 | 0.81 |
| | Thermal$_{Grey}$ | $30x224^2$ | $10.15\times3\times10$ | 81.2 | 0.80 |
| | Thermal$_{Color}$ | $30x224^2$ | $30.45\times3\times10$ | **83.4** | **0.84** |
| | Spiking 10k rate | $100x200^2$ | $21.91\times3\times10$ | 79.9 | 0.79 |
| | Spiking 1k rate | $100x200^2$ | $21.91\times3\times10$ | *69.4* | *0.69* |
| **UniFormer_B** *# Param = 50M* | RGB$_{Grey}$ | $50x224^2$ | $101.04\times1\times4$ | *83.8* | *0.82* |
| | RGB$_{Color}$ | $50x224^2$ | $303.13\times1\times4$ | 85.2 | 0.85 |
| | Thermal$_{Grey}$ | $30x224^2$ | $60.63\times1\times4$ | 84.9 | 0.85 |
| | Thermal$_{Color}$ | $30x224^2$ | $181.88\times1\times4$ | **85.7** | **0.87** |
| | Spiking 10k rate | $100x200^2$ | $77.92\times1\times4$ | 84.7 | 0.84 |
| | Spiking 1k rate | $100x200^2$ | $77.92\times1\times4$ | *74.4* | *0.74* |

Table 4: Results for different experiments Spiking data using SNN direct training and hybrid models.

| Category | Model | Data | T | Accuracy (%) | F1 Score |
|---|---|---|---|---|---|
| SNN Direct Train | STS-ResNet (Samadzadeh et al., 2020) | Spiking 10k | | 15.62 | 0.16 |
| | MS-ResNet (Hu et al., 2024) | Spiking 10k | | 50.35 | 0.50 |
| | TET-ResNet (Deng et al., 2022) | Spiking 10k | | 58.16 | 0.59 |
| Hybrid | Respike (Xiao et al., 2024) | Spiking 10k | | 71.18 | 0.72 |
| | | RGB + Spiking 10k | | 82.67 | 0.82 |
| | | Thermal + Spiking 10k | | 87.54 | 0.87 |

cially in color settings, demonstrating its ability to capture more complex features, albeit at a higher computational cost.

Directly training SNNs for video classification remains a significant challenge, with performance lagging behind ANNs by over 30%, as reported in (Xiao et al., 2024) and shown in Table 4. To address these limitations, hybrid models have been proposed (Xiao et al., 2024) to balance the trade-off between accuracy and energy efficiency through cross-attention fusion between ANN and SNN models. The results in Table 4 demonstrate that hybrid models effectively achieve this balance by leveraging spiking data in SNNs alongside RGB or thermal data in ANNs, significantly improving performance through cross-modal fusion. However, our findings also highlight the need for further advancements in SNN training, as current hybrid models remain incompatible with energy-efficient neuromorphic hardware, limiting their practical deployment.

Table 5 shows that while Spiking 10k achieves higher accuracy on the ANN, Spiking 1k results in competitive accuracy at lower latencies during SNN inference (particularly between $T = [128, 512]$). This can be attributed to temporal compression during ANN training, where both datasets are rate encoded into 100 frames, thus spiking 10k dataset has a finer temporal resolution of 0.01, while spiking 1k has a coarser resolution of 0.1. These results highlight a trade-off between temporal resolution and efficiency, with Spiking 1k excelling in low-latency, efficiency-critical applications. More experimental and qualitative results are in Appendix A.1 and A.5, respectively.

# 6 CHALLENGES, LIMITATION AND FUTURE WORK

## 6.1 VIDEO ACTION RECOGNITION CLASSIFICATION

We believe that SPACT18 can be exploited by the SNN's research community as benchmark to accelerate video understanding tasks, for efficient deployment on realworld applications.

**Direct SNN Training:** Our dataset offers rich temporal information, with raw spiking data containing 100k time steps per sample. Direct training on such large-scale data is computationally intensive, particularly for SNNs, as their training computational complexity scales with the number of time steps. This makes SPACT18 (and its compressed versions) a challenge for developing specialized algorithms tailored for direct SNN training and evaluation.

Table 5: MC3 results for ANN-SNN conversion

|  | ANN | T=16 | T=32 | T=64 | T=128 | T=256 | T=512 | T=1024 | T=2048 |
|---|---|---|---|---|---|---|---|---|---|
| Spiking 10k rate | 75.52 | 8.68 | 12.33 | 15.10 | 18.92 | 21.53 | 35.07 | 58.85 | 71.35 |
| Spiking 1k rate | 69.64 | 11.11 | 15.67 | 29.17 | 48.81 | 59.92 | 64.88 | 68.45 | 69.05 |

**ANN-SNN Conversion:** Although ANN-SNN conversion methods have made significant advancements for image classification models (Wang et al., 2023a; Bu et al., 2023; Wu et al., 2024; Jiang et al., 2023), applying these techniques to video models remains challenging. This is mainly due to the custom layers in video models whose conversion to SNN is still not well understood, and to the increased depth of these models Feichtenhofer (2020)Li et al. (2022), which consequently leads to high conversion approximation error in SNNs. Although, MC3 achieves a high accuracy, this comes at a cost of high latency (hence high energy consumption), as shown in the Table 5. This highlights the need for more efficient ANN-SNN conversion methods specific to video models.

**Multimodal Action classification:** Multimodal research in SNNs is still in its early stages, facing significant challenges compared to ANNs, particularly in integrating data like RGB and thermal camera inputs with event-driven spikes from sensors such as spike cameras (Dai et al., 2024; Rathi & Roy, 2021). While most SNN research has focused on single-modality event-driven data, the synchronization and fusion of diverse modalities, along with the development of learning algorithms to handle such multimodal inputs, remain key obstacles (Bjorndahl et al., 2024). The SNN community can benefit from multimodal datasets that include RGB, thermal, and spike camera data, enabling the creation of energy-efficient models for real-time, low-power applications like surveillance, robotics, and autonomous systems (Safa et al., 2023; Wang et al., 2023b). These datasets allow researchers to explore fusion, enhance action recognition, improve cross-modal learning, and benchmark neuromorphic hardware, ultimately advancing the use of SNNs in dynamic environments.

## 6.2 LOW LEVEL VISION APPLICATIONS

In addition to the main task of video action classification, our dataset can be used in other tasks as:

**Reconstruction:** Recently, significant research has focused on spike camera reconstruction for high-speed moving objects, with most datasets presented specifically to the reconstruction methods being developed (Zhao et al., 2020; 2021; 2024; Zhang et al., 2023b; Chen et al., 2022). Additionally, spike cameras typically require sunlight for optimal performance; however, our dataset was collected indoors under artificial lighting, presenting a new challenge. This unique setting also provides an opportunity to benchmark reconstruction algorithms under less ideal lighting conditions.

**Compression:** We have proposed a new compression algorithm to reduce the temporal resolution of raw spiking data, enabling more efficient training on smaller datasets. However, more efficient algorithms could be devloped to extract features from the original raw data and compress it into a smaller spiking representation with minimal loss of the rich temporal information inherent in the original dataset. Furthermore, our dataset can also be used to evaluate compression algorithms for efficient storage and transmission of spike camera data (Feng et al., 2023).

## 7 CONCLUSION

This paper introduces SPACT18, the first spiking VAR benchmark dataset using spike cameras synchronized with RGB and thermal modalities, advancing spiking video understanding. Key findings show spiking data achieves competitive performance with compressed 10k (rate), while compressed 1k with degraded accuracy highlights the trade-offs of extreme compression. Thermal data outperformed other modalities, achieving 85.7% accuracy with UniFormer, demonstrating the potential of multimodal fusion. In contrast, direct spiking training and ANN-SNN conversion remain challenging due to low accuracy, high latency and computational complexity. Although, Hybrid approaches like Respike excel in cross-modal integration underscoring the trade-off between accuracy and energy efficiency. SPACT18 lays a foundation for energy-efficient models, with future work focusing on optimized SNN training, improved ANN-SNN conversion, and multimodal integration for practical applications.

## 8 ETHICS STATEMENT

All necessary approvals and informed consents were obtained from the participants, ensuring that the dataset will be publicly available for research purposes. This study adheres to ethical guidelines, and no actions were taken that could compromise the privacy, safety, or well-being of the volunteer subjects. Additionally, the dataset complies with all applicable ethical standards and privacy regulations to protect the participants' identities and data.

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

## A  Appendix

### A.1  Detailed Experimental Results

This table presents a comprehensive comparison of various ANN models evaluated across different input modalities, including RGB, Thermal, and Spiking data. It details the computational cost in terms of FLOPs, alongside the accuracy and F1 scores achieved for each configuration. The models analyzed include X3D Feichtenhofer (2020), C2D Tran et al. (2018c), I3D Carreira & Zisserman (2017b), SlowFast Feichtenhofer et al. (2019a), and UniFormer Li et al. (2022), each offering a distinct trade-off between model complexity and performance.

The results show a consistent improvement in accuracy when using color inputs compared to greyscale inputs across different models. Thermal data also tends to yield superior performance in comparison to RGB for some models, indicating its potential value for specific tasks. The table highlights the differences between lightweight and more complex architectures, with models like X3D and UniFormer achieving competitive accuracy and F1 scores with significantly fewer parameters and lower FLOPs, making them more suitable for resource-constrained environments.

### A.2  Comparison with other datasets

The literature extensively uses multi-modalities for video action recognition, including RGB, thermal, depth, and IMU data. Additionally, some datasets have employed event cameras. A key differentiator of our work is including spiking data, which adds significant value.

Table 6 highlights the key features of widely used action recognition datasets and emphasizes the unique contributions of SPACT18. Unlike other datasets, SPACT18 integrates spiking data with RGB and thermal modalities, providing a comprehensive multi-modal framework for video understanding. The inclusion of a thermal camera is particularly important as it captures motion effectively under low lighting and low frame rates, complementing RGB and spike data. Synchronizing these modalities enhances the dataset's diversity and robustness, allowing for broader applications and meaningful performance comparisons. Furthermore, SPACT18 provides raw spiking data from native spike camera recordings, offering researchers the opportunity to optimize SNNs specifically for video understanding tasks. With its high-resolution data, diverse set of 18 activity classes, and multi-modal design, SPACT18 represents a notable advancement in action recognition and spiking research, driving innovation in energy-efficient video processing technologies.

The table 7 compares several action recognition datasets, highlighting features like data sources, modalities, resolutions, and class/sample counts. The SPACT18 dataset offers diverse modalities (Spike, RGB, Thermal), unlike traditional datasets (e.g., HMDB-51, UCF-101, Kinetics-400) that use only RGB data. SPACT18's higher-resolution RGB and Thermal videos enable detailed feature extraction, and its balanced and high samples per class (approximately 176) support robust training. Overall, SPACT18 bridges the gap between conventional video recognition datasets and multi-modal action recognition, enhancing understanding of complex human activities.

### A.3  Training Hyper-Parameters and Hardware Specifications

Table 8 presents the technical specifications of thermal, RGB, and spike camera. Table 9 reports the training hyper-parameters used across the three datasets modalities for all models, X3d, Uniformer, and MC3.

### A.4  Compression algorithm

We provide the proof of the main lemma here. Recall that by a firing rate of a spike train over an interval, we mean it's average spike output over the interval. By an interval we mean a sequence of consecutive time steps. By a limit spike rate of a spike train $s = [s[1], \dots, s[t], \dots]$ we mean the value (if it exists)

$$\lim_{t \to \infty} \frac{\sum_{i=1}^{t} s[i]}{t}.$$

| Model | Data | Input Size | FLOPs (G) | Accuracy% | F1 Score |
|---|---|---|---|---|---|
| **X3D_S**
*# Param = 3M* | RGB$_{Grey}$ | 50x224$^2$ | 7.46×3×10 | 75.4 | 0.76 |
| | RGB$_{Color}$ | 50x224$^2$ | 22.31×3×10 | 76.8 | 0.77 |
| | Thermal$_{Grey}$ | 30x224$^2$ | 4.47×3×10 | 77.9 | 0.79 |
| | Thermal$_{Color}$ | 30x224$^2$ | 13.41×3×10 | **80.6** | **0.80** |
| | Spiking 10k rate | 100x200$^2$ | 11.88×3×10 | 80.2 | 0.80 |
| | Spiking 1k rate | 100x200$^2$ | 11.88×3×10 | *70.4* | *0.70* |
| **X3D_M**
*# Param = 3M* | RGB$_{Grey}$ | 50x224$^2$ | 13.72×3×10 | 78.7 | *0.76* |
| | RGB$_{Color}$ | 50x224$^2$ | 41.19×3×10 | 80.9 | 0.81 |
| | Thermal$_{Grey}$ | 30x224$^2$ | 10.15×3×10 | 81.2 | 0.80 |
| | Thermal$_{Color}$ | 30x224$^2$ | 30.45×3×10 | **83.4** | **0.84** |
| | Spiking 10k rate | 100x200$^2$ | 21.91×3×10 | 79.9 | 0.79 |
| | Spiking 1k rate | 100x200$^2$ | 21.91×3×10 | *69.4* | *0.69* |
| **C2D**
*# Param = 24M* | RGB$_{Grey}$ | 50x224$^2$ | 53.02×3×10 | 71.0 | 0.71 |
| | RGB$_{3\ Channels}$ | 50x224$^2$ | 159.07×3×10 | 72.6 | 0.72 |
| | Thermal$_{Grey}$ | 30x224$^2$ | 31.81×3×10 | 73.2 | 0.73 |
| | Thermal$_{3\ Channels}$ | 30x224$^2$ | 95.43×3×10 | **74.8** | **0.74** |
| | Spiking 10k rate | 100x200$^2$ | 84.84×3×10 | 72.8 | 0.72 |
| | Spiking 1k rate | 100x200$^2$ | 84.84×3×10 | *67.4* | *0.67* |
| **I3D**
*# Param = 28M* | RGB$_{Grey}$ | 50x224$^2$ | 76.67×3×10 | 73.5 | 0.73 |
| | RGB$_{3\ Channels}$ | 50x224$^2$ | 230.02×3×10 | 75.0 | 0.75 |
| | Thermal$_{Grey}$ | 30x224$^2$ | 46.00×3×10 | 75.7 | 0.76 |
| | Thermal$_{3\ Channels}$ | 30x224$^2$ | 138.00×3×10 | **77.6** | **0.77** |
| | Spiking 10k rate | 100x200$^2$ | 122.97×3×10 | 76.1 | 0.76 |
| | Spiking 1k rate | 100x200$^2$ | 122.97×3×10 | *69.8* | *0.69* |
| **UniFormer_S**
*# Param = 21M* | RGB$_{Grey}$ | 50x224$^2$ | 22.15×1×4 | *80.6* | 0.82 |
| | RGB$_{Color}$ | 50x224$^2$ | 66.44×1×4 | 82.5 | 0.83 |
| | Thermal$_{Grey}$ | 30x224$^2$ | 13.39×1×4 | 83.0 | 0.83 |
| | Thermal$_{Color}$ | 30x224$^2$ | 40.16×1×4 | **84.2** | **0.85** |
| | Spiking 10k rate | 100x200$^2$ | 35.52×1×4 | 82.5 | 0.82 |
| | Spiking 1k rate | 100x200$^2$ | 35.52×1×4 | *73.2* | *0.72* |
| **UniFormer_B**
*# Param = 50M* | RGB$_{Grey}$ | 50x224$^2$ | 101.04×1×4 | *83.8* | *0.82* |
| | RGB$_{Color}$ | 50x224$^2$ | 303.13×1×4 | 85.2 | 0.85 |
| | Thermal$_{Grey}$ | 30x224$^2$ | 60.63×1×4 | 84.9 | 0.85 |
| | Thermal$_{Color}$ | 30x224$^2$ | 181.88×1×4 | **85.7** | **0.87** |
| | Spiking 10k rate | 100x200$^2$ | 77.92×1×4 | 84.7 | 0.84 |
| | Spiking 1k rate | 100x200$^2$ | 77.92×1×4 | *74.4* | *0.74* |
| **SlowFast**
*# Param = 35M* | RGB$_{Grey}$ | 50x224$^2$ | 97.46×3×10 | 76.7 | 0.76 |
| | RGB$_{3\ Channels}$ | 50x224$^2$ | 292.37×3×10 | 78.3 | 0.78 |
| | Thermal$_{Grey}$ | 30x224$^2$ | 58.47×3×10 | 79.1 | 0.79 |
| | Thermal$_{3\ Channels}$ | 30x224$^2$ | 175.41×3×10 | **81.2** | **0.81** |
| | Spiking 10k rate | 100x200$^2$ | 156.06×3×10 | 80.7 | 0.80 |
| | Spiking 1k rate | 100x200$^2$ | 156.06×3×10 | *71.5* | *0.71* |

**Lemma 1.** *We keep the notation as above and suppose that $s$ is a spike train obtained when an IF neuron receives constant nonnegative input $c$ at each time step, and process it according to equations equation 2. Then, for any $d$ as above, the spike train $s'$ obtained through compression and spike train $s$ will have the same limit spiking rate (which is $c$).*

*Proof.* Since the input to the neuron is constant, the exact number of firing during an interval of the form $[1, \ldots, t]$ is given by $\lfloor t \cdot c \rfloor$. The firing rate over the same interval is then

$$\frac{\lfloor t \cdot c \rfloor}{t}. \tag{3}$$

On the other side, for $t = d$, the expression 3 is exactly what the IF neuron will receive during the compression in the fist time step. After $l$ steps of compression, the total input to the neuron used for

Table 6: Comparison of Event-Based Action Recognition Datasets

| Dataset Name | Year | Sensor(s) | Resolution | Object | Scale | Classes | Scale / Classes Ratio | Subjects | Real-World | Duration | Modalities | Static |
|---|---|---|---|---|---|---|---|---|---|---|---|---|
| ASLAN-DVS (Bi et al., 2020) | 2011 | DAVIS240c | 240×180 | Action | 3,697 | 432 | ~9 | - | ✗ | - | DVS | ✗ |
| CIFAR10-DVS (Li et al., 2017) | 2017 | DAVIS128 | 128×128 | Image | 10,000 | 10 | ~1,000 | - | ✗ | 1.2s | DVS | ✗ |
| DvsGesture (Amir et al., 2021) | 2017 | DAVIS128 | 128×128 | Action | 1,342 | 11 | ~122 | 29 | ✓ | ~6s | DVS | ✗ |
| ASL-DVS (Bi et al., 2020) | 2019 | DAVIS240 | 240×180 | Hand | 100,800 | 24 | ~4,200 | 5 | ✓ | ~0.1s | DVS | ✗ |
| PAF (Miao et al., 2019) | 2019 | DAVIS346 | 346×260 | Action | 450 | 10 | ~45 | 10 | ✓ | ~5s | DVS | ✗ |
| PAFBenchmark (Miao et al., 2019) | 2019 | DAVIS346 | 346×260 | Action | 642 | 3 | ~214 | - | ✓ | - | DVS | ✗ |
| HMDB-DVS (Bi et al., 2020) | 2019 | DAVIS240c | 240×180 | Action | 6,766 | 51 | ~133 | - | ✗ | 19s | DVS | ✗ |
| UCF-DVS (Bi et al., 2020) | 2019 | DAVIS240c | 240×180 | Action | 13,320 | 101 | ~132 | - | ✗ | 25s | DVS | ✗ |
| DailyAction (Liu et al., 2021) | 2021 | DAVIS346 | 346×260 | Action | 1,440 | 12 | ~120 | 15 | ✓ | ~5s | DVS | ✗ |
| HARDVS (Wang et al., 2022a) | 2022 | DAVIS346 | 346×260 | Action | 107,646 | 300 | ~359 | 5 | ✓ | ~5s | DVS | ✗ |
| $THU^{E-ACT}-50-CHL$ (Gao et al., 2023) | 2023 | DAVIS346 | 346×260 | Action | 2,330 | 50 | ~47 | 18 | ✓ | 2-5s | DVS | ✗ |
| Bullying10K (Dong et al., 2023) | 2023 | DAVIS346 | 346×260 | Action | 10,000 | 10 | ~1,000 | 25 | ✓ | 2-20s | DVS | ✗ |
| DailyDVS-200 (Wang et al., 2024a) | 2024 | DVXplorer Lite | 320×240 | Action | 22,046 | 200 | ~110 | 47 | ✓ | 1-20s | DVS + RGB | ✓ (RGB Only) |
| SPACT18 | 2024 | Spike, RGB, Thermal | 250×400 (Spike), 1920×1080 (RGB), 1440×1080 (Thermal) | Action | 3,168 | 18 | ~176 | 44 | ✓ | ~5s | Spike + RGB + Thermal | ✓ (All Modalities) |

Table 7: Comparison of Action Recognition Datasets

| Feature | HMDB-51 | UCF-101 | Kinetics-400 | DVS128 Gesture | SPACT18 |
|---|---|---|---|---|---|
| Data Sources | Movies, web videos | YouTube videos | YouTube videos | DVS recordings | Human Subjects |
| Modality | RGB | RGB | RGB | Event-based | Spike, RGB, Thermal |
| Resolution | Varies (low-res) | 320×240 | 340×256 (average) | 128×128 | 250×400 (Spike) 1920×1080 (RGB) 1440×1080 (Thermal) |
| Number of Classes | 51 | 101 | 400 | 11 | 18 |
| Number of Samples | 6,766 | 13,320 | 306,245 | 1,342 | 3,168 |
| Avg. Duration per Sample | Varies | Varies | ~10 seconds | ~6 seconds | ~5 seconds |
| Samples per Class | ~133 | ~132 | ~765 | ~122 | ~176 |

compression will be

$$\frac{\sum_{i=0}^{l-1}\sum_{t\in I(i)} s[t]}{d} = \frac{\sum_{t=1}^{l\cdot d} s[t]}{d} = \frac{\lfloor ld\cdot c\rfloor}{d},$$

and the firing rate of the compressed spike train over the interval $[1,\dots,l]$ will be

$$\frac{1}{l}\left\lfloor \frac{\lfloor ld\cdot c\rfloor}{d}\right\rfloor. \tag{4}$$

The result follows upon comparing the equations 3 and 4, and taking the limits $t\to\infty$ and $l\to\infty$, respectively. □

The previous results does not have any prior assumptions on the constant input $c$, except it being nonnegative (negative input $c$ will yield zero spikes in both original and compressed sequence). However, if we assume that $c$ is of the form $\frac{p}{q}$, where $p,q$ are integers and $p>0, q\neq 0$, we can say more. Namely, taking $t=q$ in equation 3, yields that the firing rate of the original spike train is $\frac{p}{q}$, which is the limit firing rate too.

Table 8: Technical specifications for each camera used in data collection.

|  | Thermal Camera | RGB Camera | Spike camera |
|---|---|---|---|
| Resolution | $1440 \times 1080$ | $1920 \times 1080$ | $250 \times 400$ |
| Temperature Range | $-20°C$ to $120°C$ | - | - |
| Frame Rate | 8.7 Hz | 60 Hz | 20,000 Hz |
| Compatibility | Android (USB-C) | iOS(iPhone 14 Pro) | Windows (Laptop) |
| Manufacturer | FLIR ONE Pro | Apple | Spike Camera-001T-Gen2 |

Table 9: Training key hyperparameters for Thermal, RGB, and Spiking Data.

|  | **Thermal** | **RGB** | **Spiking** |
|---|---|---|---|
| **Number epochs** | 100 | 100 | 100 |
| **Mini batch size** | 8 | 8 | 8 |
| **Learning Rate** | $10^{-2}$ | $10^{-2}$ | $10^{-3}$ |
| **Optimizer** | Adam | Adam | AdamW |
| **Rate Scheduler** | StepLR | StepLR | StepLR |
| **Weight Decay** | $10^{-3}$ | $10^{-3}$ | 0.1 |
| **Loss Function** | Cross-Entropy | Cross-Entropy | Cross-Entropy |

## A.5 QUALITATIVE RESULTS

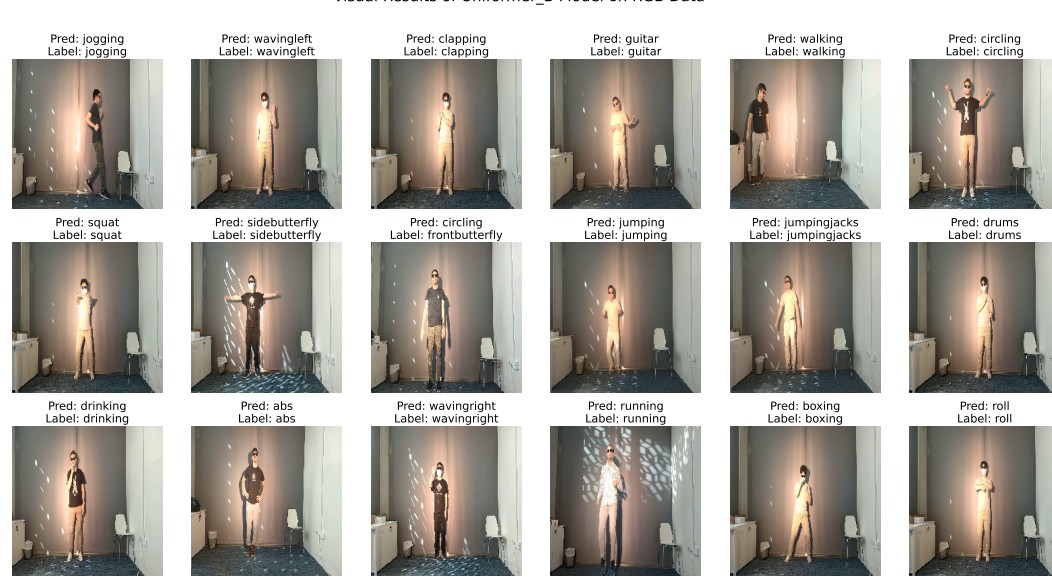

Figure 7: The figure shows visual results from the $Uniformer_B$ model applied to various activities recorded with RGB camera. Each image displays the predicted activity (Pred) and the ground truth label (Label) for comparison.

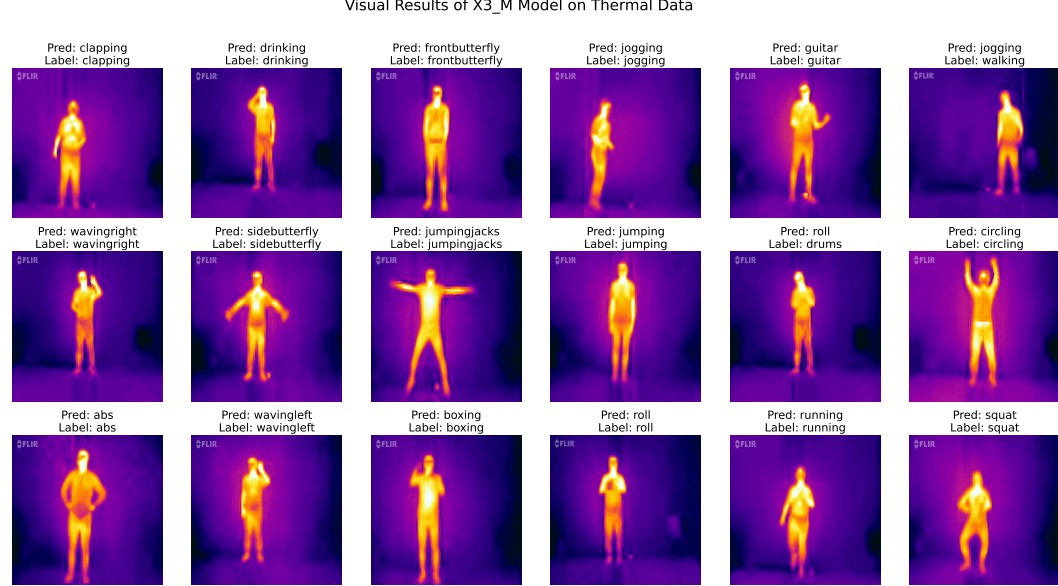

Figure 8: The figure shows visual results from the $X3D_M$ model applied to various activities recorded with Thermal camera. Each image displays the predicted activity (Pred) and the ground truth label (Label) for comparison.

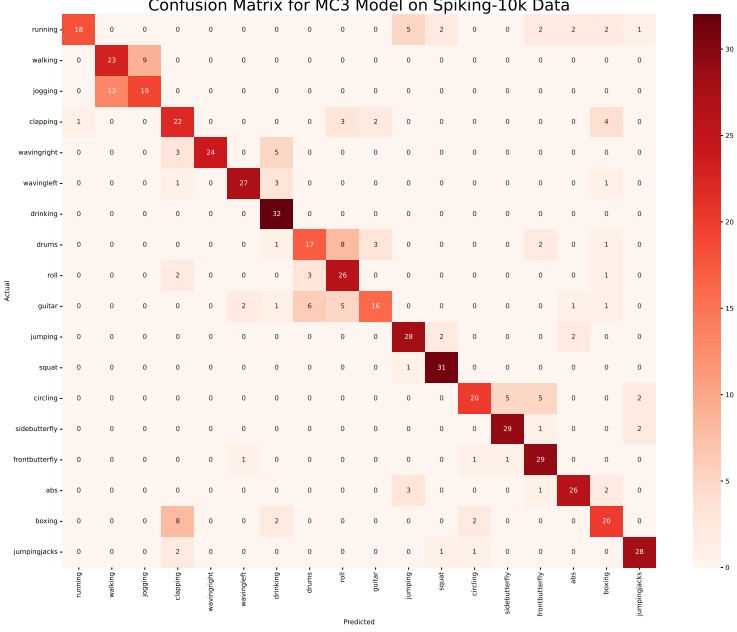

Figure 9: Confusion Matrix for $MC3$ baseline model on Spiking-10k Data: The matrix illustrates the model's performance in classifying various activities, with strong diagonal values indicating accurate predictions and some off-diagonal misclassifications, particularly in similar activities as 'walking' and 'jogging'.

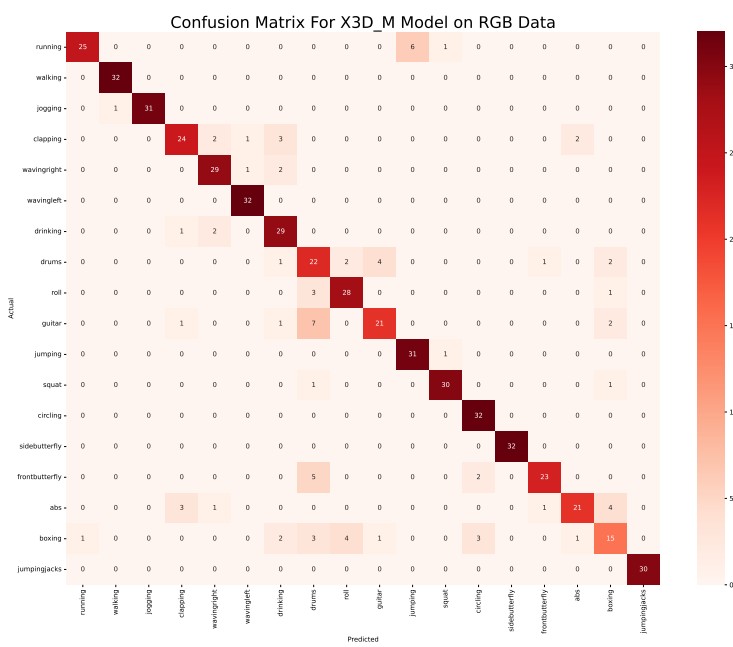

Figure 10: Confusion Matrix for $X3D_M$ Model on RGB Data. It shows strong performance in activity classification, with most predictions aligning well with actual labels, though some confusion exists, particularly in close activities like 'drums' and 'guitar.'

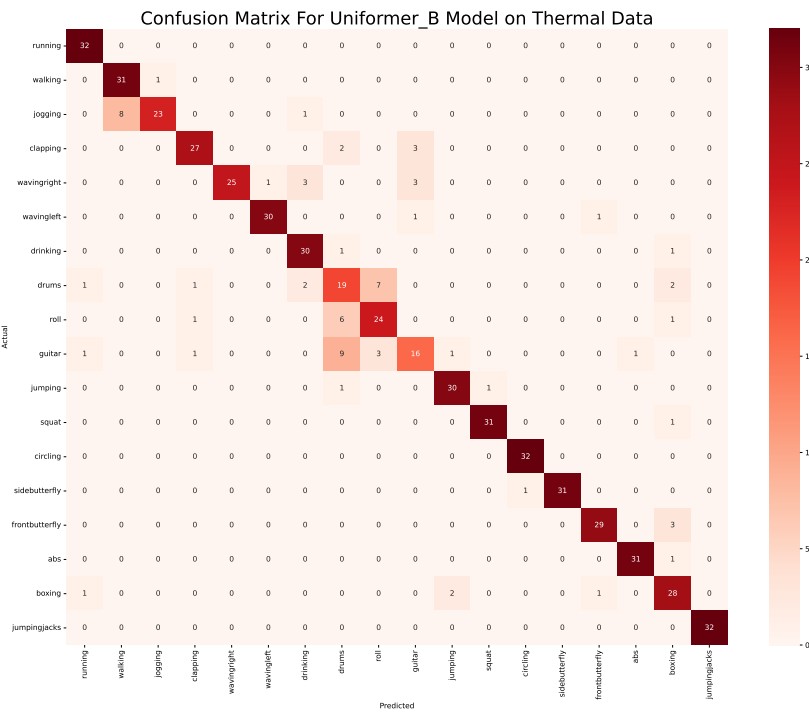

Figure 11: Confusion Matrix for $Uniformer_B$ model on Thermal Data: This matrix demonstrates the model's best overall performance, with most activities being accurately classified, particularly for 'running,' 'walking,' and 'jumping,' though minor confusion is observed in activities that are not clear in thermal imaging, like 'drums' and 'guitar.'

