# OpenReview forum: "SPACT18: Spiking Human Action Recognition Benchmark Dataset with Complementary RGB and Thermal Modalities"
_ICLR.cc/2025/Conference — Submitted to ICLR 2025_

### Official Review · Reviewer_YfhY · 2024-10-28

**Soundness:** 1
**Presentation:** 2
**Contribution:** 1
**Rating:** 3
**Confidence:** 5

**Summary:**

This paper introduces SPACT18, a novel video action recognition (VAR) dataset captured using a spike camera, alongside synchronized RGB and thermal modalities.

**Strengths:**

Introducing a spike camera-based dataset for video action recognition is a novel contribution. The dataset could become a valuable resource for researchers exploring energy-efficient spiking models for real-time applications, particularly in edge devices or autonomous systems.

**Weaknesses:**

1. The experimental setup and analysis are not sufficiently thorough.  The dataset is aimed at benchmarking Spiking Neural Networks (SNNs) for action recognition tasks, but the paper does not include any direct evaluation using SNN-based methods. Since the core claim is that this dataset enables effective benchmarking for SNNs, its advantages in temporal processing remain unproven without such an evaluation. Including results from SNN-specific architectures or methodologies (eg. directly trained SNN models and ANN-SNN conversion methods) would strengthen the paper’s contribution to the field.

2. The dataset's scale and diversity are limited compared to established RGB video action recognition benchmarks such as HMDB-51, UCF-101, and Kinetics-400, which feature far more action classes (51, 101, and 400, respectively). SPACT18 includes only 18 action classes and a significantly lower number of training clips. This restricts its utility for developing models that generalize well across diverse action recognition tasks, limiting the dataset’s value for broader use in the community.

3. Overclaiming on Spike Camera superiority. The paper claims that spike cameras offer significant advantages over event cameras. However, this claim is not supported by sufficient evidence or direct comparisons. To substantiate such claims, the authors should provide side-by-side evaluations of spike and event camera performance, particularly in relevant tasks like video action recognition, or refer to studies that have explored these comparisons in depth.

4. The paper overstates the complexity of action recognition relative to tasks such as optical flow estimation, motion estimation, depth estimation, or image reconstruction [L133-138]. Action recognition, being a classification task, is often less complex than these tasks, which involve per-pixel regression and dense predictions. It would be more accurate to frame action recognition as an important but distinct problem, rather than more complex, to avoid misrepresentation.

**Questions:**

Could you clarify the advantages of using spike cameras over event cameras, RGB cameras, and thermal data? From Table 4, it appears that performance using spike data is consistently lower, and Figure 6 shows that spike data is quite blurry, which may make it harder for models to learn meaningful dynamic information. How do you address these limitations, and in what situations does spike data excel over other modalities?

---

> ### Author Response · Authors · 2024-11-20
> **Thank you for your review**
>
> Dear Reviewer,
>
> Thank you for your review and valuable feedback. We will address the weaknesses and questions you raised below,
>
> > W1: The experimental setup and analysis are not sufficiently thorough......
>
>
> ANN-SNN conversion aims to approximate the original ANN model, but it requires the activation function to be ReLU, which is uncommon in ANN-based video classification models. We attempted to replace the Sigmoid and SiLU activation functions in X3D models with ReLU. While training the modified model resulted in less than a 1\% drop in accuracy, the ANN-SNN conversion produced random guesses, even after testing multiple ANN-SNN conversion techniques [1,2].
>
> This issue arises because X3D is a very deep model, and the error propagates through the layers. By the final layer, the classification accuracy dropped to  5\%, even with a high number of timesteps +1000 time step. While we provided a conversion for MC3, as noted in the paper, it still requires high timesteps. This underscores the need for ANN-SNN conversion techniques specifically designed for video classification tasks.
>
> On the other hand, directly training SNNs for video classification remains challenging, with performance significantly lagging behind ANNs, by over 30\%, as noted in [6]. Following your suggestion, we conducted additional experiments with existing SNN-based models [3,4,5] and the ReSpike hybrid model [6]. The results, summarized below, show that hybrid models effectively balance energy efficiency and accuracy. Notably, Feeding spiking data to SNNs and RGB or thermal data to ANNs, combined with cross-attention fusion, improved performance. However, our findings highlight the need for further efforts to enhance SNN training for video classification, such that Hybrid models are not compatible with energy-efficient neuromorphic hardware.
>
> ### Model Accuracy for SNN and Hybrid Models
>
> | **Category**        | **Model**         | **Data**              | **Accuracy (%)** |
> |----------------------|-------------------|-----------------------|------------------|
> | **SNN Direct Train** | STS-ResNet [3]  | Spiking 10k           | 15.62            |
> |                      | MS-ResNet [4]   | Spiking 10k           | 50.35            |
> |                      | TET-ResNet [5]  | Spiking 10k           | 58.16            |
> | **Hybrid**           | Respike [6]     | Spiking 10k           | 71.18            |
> |                      |                  | RGB + Spiking 10k     | 82.67            |
> |                      |                  | Thermal + Spiking 10k | 87.54            |
>
>
> > W2: The dataset's scale and diversity are limited compared to established RGB video......
>
> We acknowledge the reviewer's observation regarding the scale and diversity of SPACT18 compared to benchmarks like HMDB-51, UCF-101, and Kinetics-400. While SPACT18 includes fewer action classes (18) and a smaller number of training clips, it is important to note that our dataset pioneers the use of spike cameras for action recognition. This novel modality inherently differs from RGB datasets and introduces unique challenges and opportunities for research in neuromorphic computing and multimodal fusion. In addition to that, we want to highlight the problem of storage and high volume of data, which is the storage for raw spike data was ~95GB per subject. Notably,  DVS-Gesture [7] when it was proposed for the DVS sensor was only 11 classes with 29 subjects, however, this work was a valuable benchmark for SNN models and recently there is a dataset with 200 classes [8]. We believe that SPACT18 complements existing benchmarks by addressing a gap in the study of spike camera data, rather than directly competing with them in scale. Furthermore, as the research community adopts and builds upon this dataset, its scope can be expanded through collaborative efforts and future iterations.
>
> ### Comparison of Action Recognition Datasets
>
> | **Feature**                | **HMDB-51**         | **UCF-101**         | **Kinetics-400**       | **DVS128 Gesture**     | **SPACT18 2024**                        |
> |----------------------------|---------------------|---------------------|------------------------|------------------------|------------------------------------------|
> | **Data Sources**           | Movies, web videos | YouTube videos      | YouTube videos         | DVS recordings         | Human Subjects                           |
> | **Modality**               | RGB       | RGB  | RGB   | Event-based   | Spike, RGB, Thermal |
> | **Resolution**             | Varies (low-res)   | 320×240 | 340×256 (average)      | 128×128  | 250×400 (Spike), 1920×1080 (RGB), 1440×1080 (Thermal) |
> | **Number of Classes**      | 51       | 101   | 400    | 11   | 18  |
> | **Number of Samples**      | 6,766    | 13,320    | 306,245   | 1,342   | 3,168    |
> | **Avg. Duration per Sample** | Varies  | Varies  | ~10 seconds   | ~6 seconds   | ~5 seconds |
> | **Samples per Class**      | ~133 | ~132 | ~765 | ~122 | ~176  |

---

> ### Author Response · Authors · 2024-11-20
> **Part2**
>
> > W3 Overclaiming on Spike Camera superiority. The paper claims that spike.........
>
> The main drawback of the DVS camera is that it does not capture static scenes and struggles to capture texture details of visual scenes due to their recording of only relative changes in light intensity, leading to significantly degraded visibility. In contrast, a spiking camera records the absolute light intensity which allows it to produce spikes in static or nearly static areas, effectively displaying the background, and static objects providing rich spatial resolution [9,10].
>
> >W4: The paper overstates the complexity of action recognition relative to tasks such as optical flow estimation.......
>
> We apologize for the possible confusion in our statements,  we are not overstating the complexity of video classification but we mean that most of the existing datasets using spiking cameras actually target low-level vision tasks, but there is no dataset for action classification which is the primary task for benchmarking models for video understanding. Actually, besides its usage for action video classification, our dataset is also will be a valuable resource for image reconstruction and spiking data compression tasks [11,12].
>
> **Questions**
>
> > Q1: Could you clarify the advantages of using spike cameras over event cameras, RGB cameras, and thermal data?
>
>
> Spike cameras compared to RGB and thermal offer several advantages, particularly in their ability to produce asynchronous spike outputs that are inherently compatible with Spiking Neural Networks (SNNs) and neuromorphic hardware in addition to high temporal resolution. Compared to the event camera has an advantage in responding to static scenes, and background, and effective texture reconstruction. We don't claim superiority of spike camera over other modalities but it distinct camera with a distinct type of spike data to benchmark SNN models.
>
>
> > Q2: From Table 4, it appears that performance using spike data is consistently lower, and Figure 6 shows that spike data is quite blurry, which may make it harder for models to learn meaningful dynamic information. How do you address these limitations, and in what situations does spike data excel over other modalities?
>
> The reconstruction presented in Figure 6 is intended for visualization purposes and was done using the TFP  method [13]. It is important to note that the figure shows the compressed versions of the original data, reducing 100k spikes to 10k and 1k spikes versions. This compression inevitably introduces a degree of blurriness in the visualized output, but it does not affect the model's ability to learn from the raw spiking data. Furthermore, more advanced reconstruction methods are available [14-19], and TFB is just one of the techniques employed for the visualization purpose.
>
> Additionally, The primary purpose of our paper is not to benchmark reconstruction methods but to focus on video classification as the primary application of spike data in this paper. Reconstruction is included as a downstream task to verify and visualize that the model preserves scene information, but it is not the raw input used for training. This distinction emphasizes that while reconstruction has its uses, it does not reflect the full potential of spike cameras, which excel in tasks like video classification due to their compatibility with neuromorphic hardware.
>
> Furthermore, as shown in Table 4, the accuracy of models trained on spiking data (10k) is nearly on par with those trained on thermal and RGB data. This indicates that the information necessary for effective learning is retained in the spiking data despite the compression.
>
>
> Spiking data is preferred because of its compatibility with SNNs and neuromorphic hardware, the existence of spiking data is necessary to benchmark the SNNs model for video classification. We didn't claim that necessarily spiking data will excel over thermal and RGB data in these tasks, however, we are introducing a native dataset collected using a spike camera for benchmarking SNN models. Additionally, we have introduced the other two modalities as complementary models to accelerate the research of, including but not limited to, multimodal SNNs.
>
>
> **Summery**
> We hope that we addressed all of the reviewer's concerns, and provided the required experiments. And we are so happy to provide more answers and experiments,  We kindly ask the reviewer to reconsider their low score. Thanks for your feedback to enrich our paper with your valuable feedback!

---

> ### Author Response · Authors · 2024-11-20
> **Part3| References**
>
> **References**
> [1] Tong Bu, Wei Fang, Jianhao Ding, PENGLIN DAI, Zhaofei Yu, Tiejun Huang, Optimal ANN-SNN Conversion for High-accuracy and Ultra-low-latency Spiking Neural Networks, ICLR 2022
>
> [2]Bingsen Wang, Jian Cao, Jue Chen, Shuo Feng, Yuan Wang. A New ANN-SNN Conversion Method with High Accuracy, Low Latency and Good Robustness, IJCAI 2023.
>
> [3] Ali Samadzadeh, Fatemeh Sadat Tabatabaei Far, Ali Javadi, Ahmad Nickabadi, and Morteza Haghir Chehreghani. Convolutional spiking neural networks for spatio-temporal feature extraction. Neural Processing Letters, pages 1–17, 2023.
>
> [4] Yifan Hu, Lei Deng, Yujie Wu, Man Yao, and Guoqi Li. Advancing spiking neural networks toward deep residual learning. IEEE Transactions on Neural Networks and Learning Systems, 2024.
>
> [5] Shikuang Deng, Yuhang Li, Shanghang Zhang, and Shi Gu. Temporal efficient training of spiking neural network via gradient re-weighting, ICLR 2022.
>
> [6] Shiting Xiao, Yuhang Li, Youngeun Kim, Donghyun Lee, and Priyadarshini Panda. ReSpike: Residual frames-based hybrid spiking neural networks for efficient action recognition. arXiv 2024.
>
> [7] A. Amir et al., "A Low Power, Fully Event-Based Gesture Recognition System," CVPR 2017
>
> [8] Qi Wang, Zhou Xu, Yuming Lin, Jingtao Ye, Hongsheng Li, Guangming Zhu, Syed Afaq Ali Shah, Mohammed Bennamoun, Liang Zhang. DailyDVS-200: A Comprehensive Benchmark Dataset for Event-Based Action Recognition. ECCV 2024
>
> [9] Yajing Zheng, Lingxiao Zheng, Zhaofei Yu, Boxin Shi, Yonghong Tian, and Tiejun Huang.
> High-speed image reconstruction through short-term plasticity for spiking cameras. CVPR 2021
>
> [10] Gaole Dai, Zhenyu Wang, Qinwen Xu, Ming Lu, Wen Chen, Boxin Shi, Shanghang Zhang, Tiejun Huang. SpikeNVS: Enhancing Novel View Synthesis from Blurry Images via Spike Camera. Arxiv 2024
>
> [11]  R. Zhao et al., "Boosting Spike Camera Image Reconstruction from a Perspective of Dealing with Spike Fluctuations,"  CVPR 2024
>
> [12] Kexiang Feng, Chuanmin Jia, Siwei Ma, Wen Gao. SpikeCodec: An End-to-end Learned Compression Framework for Spiking Camera. Arxiv 2023
>
> [13] Lin Zhu, Siwei Dong, Tiejun Huang, and Yonghong Tian. A retina-inspired sampling method for visual texture reconstruction. ICME 2019.
>
> [14] Yajing Zheng, Lingxiao Zheng, Zhaofei Yu, Boxin Shi, Yonghong Tian, and Tiejun Huang. High-speed image reconstruction through short-term plasticity for spiking cameras. In CVPR 2021
>
>
> [15] Lin Zhu, Siwei Dong, Jianing Li, Tiejun Huang, and Yonghong Tian. Retina-like visual image reconstruction via spiking neural model. In CVPR 2020
>
> [16] Jing Zhao, Ruiqin Xiong, Jiyu Xie, Boxin Shi, Zhaofei Yu, WenGao, andTiejun Huang. Reconstructing clear image for high-speed motion scene with a retina-inspired spike camera. IEEE TCI 2021
>
> [17] Jiyuan Zhang, Shanshan Jia, Zhaofei Yu, and Tiejun Huang. Learning temporal-ordered representation for spike streams based on discrete wavelet transforms. In AAAI,  2023
>
> [18] Shiyan Chen, Chaoteng Duan, Zhaofei Yu, Ruiqin Xiong, and Tiejun Huang. Self-supervised mutual learning for dynamic scene reconstruction of spiking camera. In IJCAI, 2022
>
>
> [19]Wenming Weng, Yueyi Zhang, and Zhiwei Xiong. Eventbased video reconstruction using transformer. In ICCV 2021

---

> ### Author Response · Authors · 2024-11-26
>
> Dear Reviewer YfhY,
>
> We would like to know if we have addressed your concerns or if you have any follow-up questions.

---

### Official Review · Reviewer_jxr8 · 2024-11-02

**Soundness:** 2
**Presentation:** 3
**Contribution:** 2
**Rating:** 6
**Confidence:** 4

**Summary:**

This paper introduces an action recognition dataset incorporating spike, RGB, and thermal modalities. Additionally, it presents a spiking data compression algorithm designed for preprocessing and compressing the dataset. The dataset is evaluated using state-of-the-art lightweight ANN and SNN models.

**Strengths:**

- The dataset is original and diverse, featuring 44 participants and 18 activities, making it a sufficiently challenging resource for validating models.
- With three modalities—spike, RGB, and thermal—the dataset is highly informative and suitable for researchers in image processing, spiking neural networks, and multimodal analysis.
- The writing is clear, and the paper is well-structured.
- The authors validate the dataset using popular architectures, demonstrating its applicability.

**Weaknesses:**

1. The paper lacks a thorough comparison with existing action recognition datasets, particularly in terms of modality, participant diversity, types of activities, sample size, and duration. Given that the primary contribution of this work is the proposed dataset, a clear comparison would help emphasize its advantages over existing datasets.
2. The paper cites only one spike-based action recognition dataset (Amir et al., 2021). However, there exist other spike-based action recognition datasets, such as (Wang et al., 2024, Liu et al., 2021, Miao et al., 2019). It would be beneficial for the authors to consider these datasets in their comparison and explore whether any other relevant action recognition datasets are available.
3. Could you provide the confusion matrix for one or several models in Table 3? Since some activities in the proposed dataset overlap with those in [Amir et al., 2021], a confusion matrix could help determine if the additional activities are sufficiently discriminative and enhance the dataset's value.
4. The authors clearly specify the recording devices and versions for the thermal and RGB data. However, the spike camera is not similarly described. Could you include details about this device for completeness?
5. I have a question regarding the proposed compression method. Specifically, what distinguishes the approach of calculating frequency first and then using IF neurons from directly employing leaky LIF neurons? Would using leaky LIF neurons potentially offer a simpler and more direct solution?

Wang et al., 2024, DailyDVS-200: A Comprehensive Benchmark Dataset for Event-Based Action Recognition, ECCV.
Liu et al., 2021, Event-based Action Recognition Using Motion Information and Spiking Neural Networks, IJCAI.
Miao et al., 2019, Neuromorphic vision datasets for pedestrian detection, action recognition, and fall detection. Frontiers in Neurorobotics.

The authors addressed some of my concerns and I raised my score.

**Questions:**

Can you address the 1st, 3rd, 4th and 5th points I mentioned in the weakness section?

---

> ### Author Response · Authors · 2024-11-20
> **Thank you for your review**
>
> Dear Reviewer,
>
> Thank you for your positive feedback and valuable suggestions and questions to enrich our work. Below, we address your questions and concerns.
>
> >W1: The paper lacks a thorough comparison with existing action......
>
>
> We compared our dataset with similar datasets collected by the event camera as shown in below Table, we will include it in the revised version.
> ### Comparison of Event-Based Action Recognition Datasets
>
> | **Dataset Name**      | **Year** | **Sensor(s)**            | **Resolution**                                  | **Object** | **Scale** | **Classes** | **Scale/Classes Ratio** | **Subjects** | **Real-World** | **Duration**   | **Modalities**          | **Static**          |
> |------------------------|----------|--------------------------|------------------------------------------------|------------|-----------|-------------|-------------------------|---------------|----------------|----------------|-------------------------|---------------------|
> | ASLAN-DVS             | 2011     | DAVIS240c                | 240×180                                        | Action     | 3,697     | 432         | ~9                      | -             | ❌             | -              | DVS                     | ❌                  |
> | CIFAR10-DVS           | 2017     | DAVIS128                 | 128×128                                        | Image      | 10,000    | 10          | ~1,000                  | -             | ❌             | 1.2s           | DVS                     | ❌                  |
> | DvsGesture            | 2017     | DAVIS128                 | 128×128                                        | Action     | 1,342     | 11          | ~122                    | 29            | ✅             | ~6s            | DVS                     | ❌                  |
> | ASL-DVS               | 2019     | DAVIS240                 | 240×180                                        | Hand       | 100,800   | 24          | ~4,200                  | 5             | ✅             | ~0.1s          | DVS                     | ❌                  |
> | PAF                   | 2019     | DAVIS346                 | 346×260                                        | Action     | 450       | 10          | ~45                     | 10            | ✅             | ~5s            | DVS                     | ❌                  |
> | PAFBenchmark          | 2019     | DAVIS346                 | 346×260                                        | Action     | 642       | 3           | ~214                    | -             | ✅             | -              | DVS                     | ❌                  |
> | HMDB-DVS              | 2019     | DAVIS240c                | 240×180                                        | Action     | 6,766     | 51          | ~133                    | -             | ❌             | 19s            | DVS                     | ❌                  |
> | UCF-DVS               | 2019     | DAVIS240c                | 240×180                                        | Action     | 13,320    | 101         | ~132                    | -             | ❌             | 25s            | DVS                     | ❌                  |
> | DailyAction           | 2021     | DAVIS346                 | 346×260                                        | Action     | 1,440     | 12          | ~120                    | 15            | ✅             | ~5s            | DVS                     | ❌                  |
> | HARDVS                | 2022     | DAVIS346                 | 346×260                                        | Action     | 107,646   | 300         | ~359                    | 5             | ✅             | ~5s            | DVS                     | ❌                  |
> | THU E-ACT-50-CHL      | 2023     | DAVIS346                 | 346×260                                        | Action     | 2,330     | 50          | ~47                     | 18            | ✅             | 2-5s           | DVS                     | ❌                  |
> | Bullying10K           | 2023     | DAVIS346                 | 346×260                                        | Action     | 10,000    | 10          | ~1,000                  | 25            | ✅             | 2-20s          | DVS                     | ❌                  |
> | DailyDVS-200          | 2024     | DVXplorer Lite           | 320×240                                        | Action     | 22,046    | 200         | ~110                    | 47            | ✅             | 1-20s          | DVS + RGB               | ✅ (RGB Only)       |
> | **SPACT18**           | 2024     | Spike, RGB, Thermal      | 250×400 (Spike), 1920×1080 (RGB), 1440×1080 (Thermal) | Action     | 3,168     | 18          | ~176                    | 44            | ✅             | ~5s            | Spike + RGB + Thermal   | ✅ (All Modalities) |

---

> > ### Author Response · Authors · 2024-11-20
> > **Part2**
> >
> > > W2: The paper cites only one spike-based action recognition dataset.........
> >
> > Thank you for highlighting the importance of citing additional works related to event recognition datasets. Our initial focus was on citing relevant works specific to spike camera datasets, as our data is the first to utilize a spike camera for action recognition. In the revised version, we will expand the related work section to be more comprehensive.
> >
> > > W3: Could you provide the confusion matrix for one or several models in Table 3?....
> >
> > Thank you for your comment. we provided 3 different confusion matrices along with quantitative results under the QUALITATIVE RESULTS section in the Appendix attached to the paper, page 17,18,19.
> >
> > > W4: The authors clearly specify the recording devices.....
> >
> >
> > We provided the table in the appendix section, page 15; here is the table also.
> >
> > ### Technical Specifications for Each Camera Used in Data Collection
> >
> > | **Specification**  | **Thermal Camera**   | **RGB Camera**         | **Spike Camera**           |
> > |---------------------|----------------------|-------------------------|----------------------------|
> > | **Resolution**      | 1440 × 1080         | 1920 × 1080            | 250 × 400                 |
> > | **Temperature Range** | -20°C to 120°C     | -                      | -                          |
> > | **Frame Rate**      | 8.7 Hz              | 60 Hz                  | 20,000 Hz                 |
> > | **Compatibility**   | Android (USB-C)     | iOS (iPhone 14 Pro)    | Windows (Laptop)          |
> > | **Manufacturer**    | FLIR ONE Pro        | Apple                  | Spike Camera-001T-Gen2    |
> >
> > > W5: I have a question regarding the proposed compression method....
> >
> > Thank you for your insightful question. Our method compresses raw spike trains by calculating the firing rate for each time segment, followed by IF neuron dynamics to generate the compressed spike train. This approach has been theoretically validated (Lemma 1) and empirically validated through reconstruction (visualization) and action classification experiments.
> >
> > The point of proposing this compression method is to reduce the latency. The original latency of 100k time steps is too long to be tackled by the current training methods for SNNs (whether ANN-SNN conversion or direct training). So, to be able to train the models, we compress, as opposed to directly deploying LIF.
> >
> > **Summary**
> >
> > We are happy to discuss any further questions the reviewer may have and kindly ask the reviewer to increase their score. Thank you for your valuable time.

---

> ### Author Response · Authors · 2024-11-26
>
> Dear Reviewer jxr8,
>
> We would like to know if we have addressed your concerns or if you have any follow-up questions.

---

### Official Review · Reviewer_jR4n · 2024-11-03

**Soundness:** 3
**Presentation:** 3
**Contribution:** 3
**Rating:** 6
**Confidence:** 3

**Summary:**

SPACT18 has developed a novel VAR dataset for fully incorporating SNNs into VAR tasks. It is commendable that the author considered multiple variables in the process of constructing the dataset. In addition, the author's introduction is complete and logically fluent, with good narrative quality. In the methodology section, the author provides a complete introduction to the data construction process, its basic data architecture, and the most important event data alignment steps. However, in the experimental section, the author only tested the effectiveness of two typical algorithms and used the ANN-SNN conversion to test the performance of the SNN algorithm, which is incomplete. In future work, a larger number of ANN algorithms should be supplemented for testing to verify the stability of the distribution of the SPACT18 dataset. In addition, algorithm testing based entirely on SNN is also necessary. In summary, this article provides a large-scale VAR dataset based on events and innovatively proposes an event alignment strategy, which is in line with the logic and innovation required for conference papers.

**Strengths:**

This article proposes SPACT18- a novel event based large-scale VAR task dataset. The dynamic response characteristics of events and the low power consumption of SNNs provide new possibilities for the development of VAR. Meanwhile, the method section innovatively proposes an alignment method for event pulses, which is a unique compression method suitable for VAR tasks. This article has made certain progress and contributions to VAR tasks.

**Weaknesses:**

The experimental part appears slightly thin. Using only two typical VAR algorithms for verification is insufficient for verifying the distribution of RGB and thermal modal data. At the same time, the validation of the SNN algorithm is still based on the ANN-SNN conversion method, which cannot fully utilize the parsing ability that SNN should have.

**Questions:**

(1) Has a detailed analysis been conducted on the impact of external factors such as age, gender, and race on the distribution of the dataset and the effectiveness of VAR;
(2) Does the architecture of the SPACT18 dataset include commonly used data collection methods (RGB/thermal mode) and analyze the differences between SPACT18 and other existing datasets;
(3) Due to the fact that video understanding typically adopts frame by frame input, i.e. still in the form of a single frame image sequence. Is there an analysis of the contribution of each frame to the video action recognition task? I believe that different cumulative states of the event stream can be equivalent to one frame of video image, and the contribution of each frame will be particularly prominent in the event camera;

---

> ### Author Response · Authors · 2024-11-20
> **Thank you for your review**
>
> Dear Reviewer,
>
> We thank you for your positive assessment of our paper and for suggesting further improvements. Please find our responses to your questions and concerns below.
>
> > Weaknesses: The experimental part appears slightly thin........
>
> We utilized X3D and UniFormer as they represent SOTA architectures for convolution-based and transformer-based models, respectively,  in action recognition and video understanding.
>
> Following your suggestion to make our evaluation more comprehensive. We have conducted more experiments using different models on both ANN and SNN direct training.  For ANN model, we have conducted more experiments on C2D, I3D, SlowFast models, the results are presented below
>
> ### Results for  ANN Models
>
> | **Model**       | **Data**                | **Accuracy %** | **F1 Score** |
> |------------------|-------------------------|----------------|--------------|
> | **C2D**         | RGB\_Grey               | 71.0           | 0.71         |
> |                 | RGB\_3 Channels         | 72.6           | 0.72         |
> |                 | Thermal\_Grey           | 73.2           | 0.73         |
> |                 | Thermal\_3 Channels     | **74.8**       | **0.74**     |
> |                 | Spiking 10k rate        | 72.8           | 0.72         |
> |                 | Spiking 1k rate         | *67.4*         | *0.67*       |
> | **I3D**         | RGB\_Grey               | 73.5           | 0.73         |
> |                 | RGB\_3 Channels         | 75.0           | 0.75         |
> |                 | Thermal\_Grey           | 75.7           | 0.76         |
> |                 | Thermal\_3 Channels     | **77.6**       | **0.77**     |
> |                 | Spiking 10k rate        | 76.1           | 0.76         |
> |                 | Spiking 1k rate         | *69.8*         | *0.69*       |
> | **SlowFast**    | RGB\_Grey               | 76.7           | 0.76         |
> |                 | RGB\_3 Channels         | 78.3           | 0.78         |
> |                 | Thermal\_Grey           | 79.1           | 0.79         |
> |                 | Thermal\_3 Channels     | **81.2**       | **0.81**     |
> |                 | Spiking 10k rate        | 80.7           | 0.80         |
> |                 | Spiking 1k rate         | *71.5*         | *0.71*       |
>
>
> On the other hand, directly training SNNs for video classification remains challenging, with performance significantly lagging behind ANNs, by over 30\%, as noted in [4]. Following your suggestion, we conducted additional experiments with existing SNN-based models [1,2,3] and the ReSpike hybrid model [4]. The results, summarized below, show that hybrid models effectively balance energy efficiency and accuracy. Notably, Feeding spiking data to SNNs and RGB or thermal data to ANNs, combined with cross-attention fusion, improved performance. However, our findings highlight the need for further efforts to enhance SNN training for video classification, such that Hybrid models are not compatible with energy-efficient neuromorphic hardware.
>
>
> ### Results  for SNN and Hybrid Models
>
> | **Category**        | **Model**         | **Data**              | **Accuracy (%)** |
> |----------------------|-------------------|-----------------------|------------------|
> | **SNN Direct Train** | STS-ResNet [1]  | Spiking 10k           | 15.62            |
> |                      | MS-ResNet [2]   | Spiking 10k           | 50.35            |
> |                      | TET-ResNet [3]  | Spiking 10k           | 58.16            |
> | **Hybrid**           | Respike [4]     | Spiking 10k           | 71.18            |
> |                      |                  | RGB + Spiking 10k     | 82.67            |
> |                      |                  | Thermal + Spiking 10k | 87.54            |
>
> **Questions**
>
> > Q1:  Has a detailed analysis been conducted on the impact of external factors such as age, gender, and race....
>
>
> Thank you for your valuable insights. We confirm that the subjects used in training and testing have no significant impact on the model's performance. Our dataset encompasses a diverse range of genders and races, with an age span of 21 to 42 years. It includes 13 different nationalities and represents both males and females. During training and testing, we employed various random splits to check the impact, which consistently yielded comparable results with a standard deviation of less than 1\%.

---

> > ### Author Response · Authors · 2024-11-20
> > **Part 2**
> >
> > >Q2: Does the architecture of the SPACT18 dataset include commonly used data collection....
> >
> > The literature has widely used multi-modalities for video action recognition datasets, including RGB, thermal, depth cameras, and other modalities such as IMUs. Additionally, some datasets have been collected using event cameras. The inclusion of spiking data sets our work apart, which is a valuable contribution. While using RGB and thermal modalities is not new, integrating these modalities with spiking data is a novel approach. Furthermore, the availability of raw spiking data represents a significant and unique contribution to the field. We compared our dataset with similar datasets collected by event camera as shown in the below Table, we will include it in the revised version.
> > ### Comparison of Event-Based Action Recognition Datasets
> >
> > | **Dataset Name**      | **Year** | **Sensor(s)**            | **Resolution**                                  | **Object** | **Scale** | **Classes** | **Scale/Classes Ratio** | **Subjects** | **Real-World** | **Duration**   | **Modalities**          | **Static**          |
> > |------------------------|----------|--------------------------|------------------------------------------------|------------|-----------|-------------|-------------------------|---------------|----------------|----------------|-------------------------|---------------------|
> > | ASLAN-DVS             | 2011     | DAVIS240c                | 240×180         | Action     | 3,697     | 432         | ~9                      | -             | ❌             | -              | DVS                     | ❌                  |
> > | CIFAR10-DVS           | 2017     | DAVIS128                 | 128×128        | Image      | 10,000    | 10          | ~1,000                  | -             | ❌             | 1.2s           | DVS                     | ❌                  |
> > | DvsGesture            | 2017     | DAVIS128                 | 128×128         | Action     | 1,342     | 11          | ~122                    | 29            | ✅             | ~6s            | DVS                     | ❌                  |
> > | ASL-DVS               | 2019     | DAVIS240                 | 240×180      | Hand       | 100,800   | 24          | ~4,200                  | 5             | ✅             | ~0.1s          | DVS                     | ❌                  |
> > | PAF                   | 2019     | DAVIS346                 | 346×260    | Action     | 450       | 10          | ~45                     | 10            | ✅             | ~5s            | DVS                     | ❌                  |
> > | PAFBenchmark          | 2019     | DAVIS346                 | 346×260      | Action     | 642       | 3           | ~214                    | -             | ✅             | -              | DVS                     | ❌                  |
> > | HMDB-DVS              | 2019     | DAVIS240c                | 240×180    | Action     | 6,766     | 51          | ~133                    | -             | ❌             | 19s            | DVS                     | ❌                  |
> > | UCF-DVS               | 2019     | DAVIS240c                | 240×180     | Action     | 13,320    | 101         | ~132                    | -             | ❌             | 25s            | DVS                     | ❌                  |
> > | DailyAction           | 2021     | DAVIS346                 | 346×260  | Action     | 1,440     | 12          | ~120                    | 15            | ✅             | ~5s            | DVS                     | ❌                  |
> > | HARDVS                | 2022     | DAVIS346                 | 346×260    | Action     | 107,646   | 300         | ~359                    | 5             | ✅             | ~5s            | DVS                     | ❌                  |
> > | THU E-ACT-50-CHL      | 2023     | DAVIS346                 | 346×260     | Action     | 2,330     | 50          | ~47                     | 18            | ✅             | 2-5s           | DVS                     | ❌                  |
> > | Bullying10K           | 2023     | DAVIS346                 | 346×260      | Action     | 10,000    | 10          | ~1,000                  | 25            | ✅             | 2-20s          | DVS                     | ❌                  |
> > | DailyDVS-200          | 2024     | DVXplorer Lite           | 320×240      | Action     | 22,046    | 200         | ~110                    | 47            | ✅             | 1-20s          | DVS + RGB               | ✅ (RGB Only)       |
> > | **SPACT18**           | 2024     | Spike, RGB, Thermal      | 250×400 (Spike), 1920×1080 (RGB), 1440×1080 (Thermal) | Action     | 3,168     | 18          | ~176                    | 44            | ✅             | ~5s            | Spike + RGB + Thermal   | ✅ (All Modalities) |

---

> ### Author Response · Authors · 2024-11-20
> **Part 3**
>
> > Q3: Due to the fact that video understanding typically adopts frame-by-frame input........
>
> Thank you for your insightful comment. In our study, we thoroughly examined the impact of individual frames and cumulative states on overall recognition performance. Specifically, we conducted ablation experiments by selectively removing or masking certain frames to evaluate their influence on the task. This included testing different frame selections and analyzing their impact, particularly for RGB and thermal videos. We found that uniform sampling across frames yielded the best results, as it minimizes information loss while avoiding redundant frames.
>
> For spiking camera has a resolution of 20k fps, which is more than sufficient to capture all necessary information. Unlike event-based cameras, spiking cameras can capture both static and dynamic events, making them less sensitive to frame removal. However, we conducted an ablation study on the effects of the compression algorithm (as detailed in the paper) by removing frames. The results showed that increased compression (i.e., more frame removal) led to visually degraded outputs and reduced model performance. This highlights the importance of preserving frames in spiking data, although the dependency is not tied to specific frames as it often is with event cameras. Again, we chose uniform sampling across frames as in RGB as it reduces the needed frames without hurting the performance.
>
> **Summary**
>
> We hope we answered all questions and weaknesses, conducted the required experiments and made the required comparisons to make the reviewer more confident about the contribution of our paper and we kindly ask the reviewer to increase their score. Thank you!
>
> **References**
>
>
>
> [1] Ali Samadzadeh, Fatemeh Sadat Tabatabaei Far, Ali Javadi, Ahmad Nickabadi, and Morteza Haghir Chehreghani. Convolutional spiking neural networks for spatio-temporal feature extraction. Neural Processing Letters, pages 1–17, 2023.
>
> [2] Yifan Hu, Lei Deng, Yujie Wu, Man Yao, and Guoqi Li. Advancing spiking neural networks toward deep residual learning. IEEE Transactions on Neural Networks and Learning Systems, 2024.
>
> [3] Shikuang Deng, Yuhang Li, Shanghang Zhang, and Shi Gu. Temporal efficient training of spiking neural network via gradient re-weighting, ICLR 2022.
>
> [4] Shiting Xiao, Yuhang Li, Youngeun Kim, Donghyun Lee, and Priyadarshini Panda. ReSpike: Residual frames-based hybrid spiking neural networks for efficient action recognition. arXiv 2024.

---

> ### Author Response · Authors · 2024-11-26
>
> Dear Reviewer jR4n,
>
> We would like to know if we have  addressed your concerns or if you have any follow-up questions.

---

### Official Review · Reviewer_dDYr · 2024-11-04

**Soundness:** 3
**Presentation:** 2
**Contribution:** 4
**Rating:** 5
**Confidence:** 5

**Summary:**

This paper proposes the first video action recognition using spike camera, alongside synchronized RGB and thermal modalities.

**Strengths:**

1. This paper is simple and easy to understand.

**Weaknesses:**

1.  The authors emphasize that the temporal resolution of spike cameras is superior to that of event cameras.  But I think that's wrong. The proof is not a hardware paper but an algorithm paper, which I don't think is a good starting point. The paper [1]  would have been able to do 3.6us  temporal resolution compared to the spike camera even in 2011.

[1] Leñero-Bardallo J A, Serrano-Gotarredona T, Linares-Barranco B. A 3.6$\mu $ s Latency Asynchronous Frame-Free Event-Driven Dynamic-Vision-Sensor[J]. IEEE Journal of Solid-State Circuits, 2011, 46(6): 1443-1455.

2. In line 60,  the thermal modality is used very suddenly, just to provide a comprehensive multimodal representation, and why this modality is important, I think, needs to be elaborated.

3. The references are out of date and need a more comprehensive review.

4. Different models should be utilized in the experiment to give more comparable results.

5. There is a lack of comparison of model experiments for SNNs that direct train.

6. Please reference these paper to set up and complete the experiment part [2],[3].

[2] Wang X, Wang S, Tang C, et al. Event stream-based visual object tracking: A high-resolution benchmark dataset and a novel baseline[C]//Proceedings of the IEEE/CVF Conference on Computer Vision and Pattern Recognition. 2024: 19248-19257.
[3] Duan Y. LED: A Large-scale Real-world Paired Dataset for Event Camera Denoising[C]//Proceedings of the IEEE/CVF Conference on Computer Vision and Pattern Recognition. 2024: 25637-25647.

**Questions:**

None

---

> ### Author Response · Authors · 2024-11-20
> **Thank you for your review**
>
> Dear Reviewer,
>
> Thank you for taking the time to read and carefully review our paper.
>
> > W1: The authors emphasize that the temporal.....
>
> Thank you for pointing paper [1] to our attention. However, although the paper [1] establishes theoretically that DVS sensors can have a very low temporal resolution, DVS sensors have one fundamental drawback compared to spiking cameras, DVS struggle to capture texture details of visual scenes. DVS camera generates events only when there is a change in pixel brightness. It does not respond to static areas of the scene and, therefore, cannot represent the background. It is sensitive exclusively to dynamic changes, such as moving objects or variations in lighting. In contrast, a spiking camera simulates neural spike activity and can continue generating pulse activity even when changes in the scene are minimal [2,3]. This allows it to produce pulses in static or nearly static areas, effectively displaying the background. The spiking camera's ability to mimic neural firing patterns enables it to capture subtle, ongoing background information, even in the absence of significant changes in the scene [[2,3].
>
> Additionally, we want to highlight that our paper presents a novel native spike dataset for action recognition using a spiking camera. Existing spiking camera datasets primarily focus on low-vision tasks, where video sequences are on a millisecond timescale, requiring high camera motion. Our dataset differs in that each sample contains approximately 100,000 frames, opening new research questions, such as how to efficiently train this high volume of spike data?  how do compress this high volume of data for transmission, processing, and extracting important features?..etc. Our dataset has a wide range of potential applications, as discussed in the last section of the paper that we enlist briefly here including but not limited to: a benchmark for SNN-Video models, multimodal-SNN, Spiking data compression, reconstruction..etc.
>
> > W2: In line 60, the thermal modality...........
>
> Thank you for your valuable suggestion and question. We will provide a detailed explanation in the revised version of the paper.
>
> In the scope SNNs, multimodal research remains relatively limited and underexplored. To address this gap, we have included these datasets to encourage the development of multimodal architectures using SNNs, thereby increasing the versatility and applicability of our dataset.
>
> The inclusion of a thermal camera is particularly important because it captures motion effectively at low frame rates, even under low lighting conditions. By synchronizing thermal cameras with RGB cameras, we aim to enhance the diversity and robustness of our dataset, enabling broader applications and better performance comparisons across modalities.
>
> > W3: The references are out of date......
>
> Thanks for your observation, The revised version will include up-to-date relevant and related work.
>
> > W4: Different models should be.....
>
> We utilized X3D and UniFormer as they represent SOTA architectures for convolution-based and transformer-based models in action recognition, respectively. However, following your suggestion, we conducted a more comprehensive set of experiments on additional ANN models as presented in Table below.
>
>
> | **Model**                 | **Data**                | **Accuracy %** | **F1 Score** |
> |----------------------------------|-------------------------|----------------|--------------|
> | **C2D**  | RGB\_Grey               | 71.0           | 0.71         |
> |                | RGB\_3 Channels         | 72.6           | 0.72         |
> |                | Thermal\_Grey           | 73.2           | 0.73         |
> |                | Thermal\_3 Channels     | **74.8**       | **0.74**     |
> |                | Spiking 10k rate        | 72.8           | 0.72         |
> |                | Spiking 1k rate         | *67.4*         | *0.67*       |
> | **I3D**    | RGB\_Grey               | 73.5           | 0.73         |
> |                | RGB\_3 Channels         | 75.0           | 0.75         |
> |                | Thermal\_Grey           | 75.7           | 0.76         |
> |                | Thermal\_3 Channels     | **77.6**       | **0.77**     |
> |                | Spiking 10k rate        | 76.1           | 0.76         |
> |                       | Spiking 1k rate         | *69.8*         | *0.69*       |
> | **SlowFast**  | RGB\_Grey               | 76.7           | 0.76         |
> |                        | RGB\_3 Channels         | 78.3           | 0.78         |
> |                        | Thermal\_Grey           | 79.1           | 0.79         |
> |                        | Thermal\_3 Channels     | **81.2**       | **0.81**     |
> |                        | Spiking 10k rate        | 80.7           | 0.80         |
> |                        | Spiking 1k rate         | *71.5*         | *0.71*       |

---

> > ### Comment · Reviewer_dDYr · 2024-11-22
> >
> > This conference can update the revised manuscript, and you can upload it.

---

> > > ### Author Response · Authors · 2024-11-24
> > >
> > > We have uploaded the revised version, did we address your previous concerns about our paper, and is there something else we could further clarify?

---

> ### Author Response · Authors · 2024-11-20
> **Part 2**
>
> > W5: There is a lack of comparison of model experiments for SNNs that direct train:
>
> Directly training SNNs for video classification remains challenging, with performance significantly lagging behind ANNs, by over 30 \%, as noted in [4]. Following your suggestion, we conducted additional experiments with existing SNN-based models [5,6,7] and the ReSpike hybrid model [4]. The results, summarized below, show that hybrid models effectively balance energy efficiency and accuracy. Notably,  Feeding spiking data to SNNs and RGB or thermal data to ANNs, combined with cross-attention fusion, improved performance. However, our findings highlight the need for further efforts to enhance SNN training for video classification, such that Hybrid models are not compatible with energy-efficient neuromorphic hardware.
>
>
>
> ### Model Accuracy for SNN and Hybrid Models
>
> | **Category**        | **Model**         | **Data**              | **Accuracy (%)** |
> |----------------------|-------------------|-----------------------|------------------|
> | **SNN Direct Train** | STS-ResNet [5]  | Spiking 10k           | 15.62            |
> |                      | MS-ResNet [6]   | Spiking 10k           | 50.35            |
> |                      | TET-ResNet [7]  | Spiking 10k           | 58.16            |
> | **Hybrid**           | Respike [4]     | Spiking 10k           | 71.18            |
> |                      |                  | RGB + Spiking 10k     | 82.67            |
> |                      |                  | Thermal + Spiking 10k | 87.54            |
>
> >W6: Please reference these papers [8,9]:
>
>
> Thanks for providing us with these valuable references to enhance our work, we will include these references in our revised paper.
>
> **Summary**
>
> We hope that we addressed all of the reviewer's concerns, and provided the required experiments. We kindly ask the reviewer to reconsiders their low score since the reviewer recognised our contribution as an Excellent. Our dataset will be a valuable resource for accelerating research in SNN video recognition and multimodal SNN.
>
> **References**
>
>
> [1] Leñero-Bardallo J A, Serrano-Gotarredona T, Linares-Barranco B. A 3.6 s Latency Asynchronous Frame-Free Event-Driven Dynamic-Vision-Sensor[J]. IEEE Journal of Solid-State Circuits, 2011, 46(6): 1443-1455.
>
> [2] Yajing Zheng, Lingxiao Zheng, Zhaofei Yu, Boxin Shi, Yonghong Tian, and Tiejun Huang.
> High-speed image reconstruction through short-term plasticity for spiking cameras. CVPR 2021.
>
> [3] Siwei Dong, Tiejun Huang, Yonghong Tian. Spike Camera and Its Coding Methods, Arxiv 2021.
>
> [4] Shiting Xiao, Yuhang Li, Youngeun Kim, Donghyun Lee, and Priyadarshini Panda. ReSpike: Residual frames-based hybrid spiking neural networks for efficient action recognition. arXiv 2024.
>
> [5] Ali Samadzadeh, Fatemeh Sadat Tabatabaei Far, Ali Javadi, Ahmad Nickabadi, and Morteza Haghir Chehreghani. Convolutional spiking neural networks for spatio-temporal feature extraction. Neural Processing Letters, pages 1–17, 2023.
>
>
> [6] Yifan Hu, Lei Deng, Yujie Wu, Man Yao, and Guoqi Li. Advancing spiking neural networks toward deep residual learning. IEEE Transactions on Neural Networks and Learning Systems, 2024.
>
> [7] Shikuang Deng, Yuhang Li, Shanghang Zhang, and Shi Gu. Temporal efficient training of spiking neural network via gradient re-weighting, ICLR 2022.
>
> [8] Wang X, Wang S, Tang C, et al. Event stream-based visual object tracking: A high-resolution benchmark dataset and a novel baseline[C]. CVPR 2024.
>
> [9] Duan Y. LED: A Large-scale Real-world Paired Dataset for Event Camera Denoising. CVPR 2024

---

> ### Author Response · Authors · 2024-11-26
>
> Dear Reviewer dDYr,
>
> We would like to know if we have addressed your concerns or if you have any follow-up questions.

---

### Meta-Review · Area_Chair_Ty6w · 2024-12-21

**Metareview:**

The paper has undergone review by four referees, with two suggesting acceptance and two others expressing reservations. Regarding the research motivation, the paper proposes using a Spike Camera for motion detection; however, the camera’s strength lies in capturing high-speed movement, not in performing well under low-light conditions. The inclusion of thermal infrared and RGB cameras could address this issue, but the paper’s focus on the Spike Camera is less convincing. Taking into account the concerns raised by the other reviewers, the decision is to reject the manuscript.

**Additional Comments On Reviewer Discussion:**

As for the research impetus, the paper introduces the use of Spike Cameras for action recognition. Although these cameras excel in capturing rapid movements, they are not optimized for performance in low-light settings. The suggestion to integrate thermal infrared and RGB cameras could potentially mitigate this shortcoming, yet the paper’s primary emphasis on the Spike Camera appears less compelling.

---

### Decision · Program_Chairs · 2025-01-22

Reject